# Visualizing conformational dynamics of proteins in solution and at the cell membrane

**Sharona E Gordon\*, Mika Munari, William N Zagotta\***

Department of Physiology and Biophysics, University of Washington, Seattle, United States

**Abstract** Conformational dynamics underlie enzyme function, yet are generally inaccessible via traditional structural approaches. FRET has the potential to measure conformational dynamics in vitro and in intact cells, but technical barriers have thus far limited its accuracy, particularly in membrane proteins. Here, we combine amber codon suppression to introduce a donor fluorescent noncanonical amino acid with a new, biocompatible approach for labeling proteins with acceptor transition metals in a method called ACCuRET (Anap Cyclen-Cu$^{2+}$ resonance energy transfer). We show that ACCuRET measures absolute distances and distance changes with high precision and accuracy using maltose binding protein as a benchmark. Using cell unroofing, we show that ACCuRET can accurately measure rearrangements of proteins in native membranes. Finally, we implement a computational method for correcting the measured distances for the distance distributions observed in proteins. ACCuRET thus provides a flexible, powerful method for measuring conformational dynamics in both soluble proteins and membrane proteins.
DOI: https://doi.org/10.7554/eLife.37248.001

**\*For correspondence:**
seg@uw.edu (SEG);
zagotta@uw.edu (WNZ)

**Competing interests:** The authors declare that no competing interests exist.

## Introduction

Structural dynamics of proteins, particularly those at cell membranes, underlie many cell-signaling events (*Henzler-Wildman and Kern, 2007*). Structural rearrangements in membrane proteins can occur in response to binding of extracellular or intracellular ligands, covalent modification (e.g. phosphorylation), changes in membrane voltage, and mechanical forces in the membrane. These rearrangements, in turn, regulate enzyme activity, open pores, and transport molecules across the membrane. In this way, membrane proteins serve as molecular transducers that mediate the communication and transport between the cell and the external world. The molecular mechanisms, however, for these transduction processes remain largely unknown.

The number of solved membrane protein structures has grown exponentially in the last 30 years, and there are now more than 750 unique structures in the protein structure database (*Stephen White Laboratory, 1998*). These structures provide high-resolution information on the position of each atom in the protein, but provide little information about the functional states, number of states, relative energies of the states, or allowed transitions. In contrast, functional measurements like electrophysiology provide high-resolution information on states and energetics, but little information on structure. In addition, electrophysiology gives an excellent readout on rearrangements in the pore domain but is a poor surrogate for probing structural rearrangements and energetics in other regions, for example agonist/antagonist binding sites. This state of affairs has left a gap in our ability to link structure and function, as we often do not know the functional state for a particular structure or the structural state(s) that underlie a particular function.

To fill this gap, a method is needed that allows us to measure the structural dynamics and energetics of membrane proteins, preferably in their native membrane environment. Many of the existing

methods for measuring structural dynamics have limitations. Recent advances in cryo-electron microscopy have greatly facilitated the measurement of multiple protein conformations, often in the same sample (*Cheng, 2015*). However, these conformational states can only be distinguished if they have substantial structural differences, and their functional state and energetics are still unknown. Magnetic resonance methods have proven useful for measuring structural dynamics but require high concentrations of purified protein, a particular challenge for membrane proteins. Finally, fluorescence resonance energy transfer (FRET) provides a sparse measurement of protein structure and can be recorded on a biological time scale (milliseconds to seconds) on small quantities of protein (even single molecules) in their native environment (*Taraska and Zagotta, 2010*; *Stryer and Haugland, 1967*). Standard FRET, however, suffers from nonspecific labeling, inappropriate distance range for intramolecular measurements, and inaccurate measurements of distances and distance changes (*Best et al., 2007*; *Schuler et al., 2005*).

Our goal was to establish and optimize a method for measuring and analyzing the structural dynamics of membrane proteins in their native membrane environment. Our approach, which we call ACCuRET (*Anap Cyclen-Cu$^{2+}$Resonance Energy Transfer*), combines together three technologies: (1) transition metal ion FRET (tmFRET) for accurately measuring interatomic distances, (2) amber codon suppression to specifically label the protein with a fluorescent, noncanonical amino acid as the FRET donor, and (3) a novel orthogonal and biocompatible method for introducing a specific, a high-affinity binding site for a metal ion that serves as the FRET acceptor.

To measure interatomic distances, we have used tmFRET between a fluorophore and a transition metal divalent cation (*Yu et al., 2013*; *Taraska et al., 2009a*; *Taraska et al., 2009b*). Transition metal cations, such as $Ni^{2+}$, $Co^{2+}$, and $Cu^{2+}$, act as non-fluorescent FRET acceptors, quenching the fluorescence of donor fluorophores with an efficiency that is highly distance dependent (*Horrocks et al., 1975*; *Latt et al., 1972*; *Richmond et al., 2000*; *Sandtner et al., 2007*; *Latt et al., 1970*). Thus, tmFRET provides a sparse, dynamic measurement of structure on minute amounts of functioning proteins in their native environment. tmFRET has a number of significant advantages over standard FRET methods for measuring rearrangements in proteins (*Taraska and Zagotta, 2010*). (1) tmFRET has a working range of about 10–20 Å, much shorter than standard FRET, and on the order of the distance between proximal structural elements in proteins. (2) tmFRET is steeply distance dependent, but has little orientation dependence because the metal ion has multiple transition dipole moments (*Selvin, 2002*). (3) The method utilizes minimal metal binding sites, making the position of the metal a more faithful representation of the position of the protein backbone. (4) Metal binding is rapid and reversible so the fraction of fluorescence quenched at a saturating concentration of metal gives the absolute FRET efficiency and therefore the distance. And (5) metals with different $R_0$ values (distance producing 50% FRET efficiency) can be chosen to tune the measurement to the distance of interest.

The power of using fluorescence to probe protein dynamics within the cell is limited by the specificity with which a protein can be labeled with a fluorophore. For measuring the structure and dynamics of a protein, it is also important to use a small fluorophore attached to the protein with a short linker (*Taraska and Zagotta, 2010*). To achieve this specific fluorescent labeling with a small fluorophore, we used amber codon suppression to introduce the fluorescent, noncanonical amino acid L-Anap (*Chatterjee et al., 2013*; *Kalstrup and Blunck, 2013*; *Lee et al., 2009*). We chose L-Anap, a derivative of prodan, because of its small size (about the same as the amino acid tryptophan) and useful spectral properties. By introducing a fluorophore as the side-chain of an amino acid, we eliminated the need for a linker whose distance from the protein backbone and flexibility reduces the accuracy and empirical sensitivity of FRET (*Best et al., 2007*; *Schuler et al., 2005*; *Taraska et al., 2009b*; *Sobakinskaya et al., 2018*). L-Anap is therefore useful as a tmFRET donor for distance measurements.

Membrane proteins are fully functional only in their native membrane environment. For many membrane proteins, studying the relevant structural rearrangements demands access to the cytoplasmic side of the protein for introduction of ligands, probes, etc. To study membrane proteins, we have recently implemented cell unroofing, an approach borrowed from the EM literature (*Heuser, 2000*), as a medium-throughput platform for measuring tmFRET from membrane proteins in native cell membranes (*Gordon et al., 2016*; *Zagotta et al., 2016*). This technique utilizes a mechanical shearing force to dislodge the dorsal surfaces of cells and remove all soluble cellular contents and organelles, leaving the ventral surfaces of the cells intact as plasma membrane sheets

containing their native lipids and membrane proteins. Cell unroofing provides access to the intracellular surface of the bilayer for application of transition metals and intracellular ligands in an environment more physiological than membrane proteins reconstituted into synthetic liposomes and much higher throughput than inside-out patch-clamp recording.

Previous studies using FRET on membrane proteins have produced uncertain results about the size of structural rearrangements, often underestimating changes in distance (*Taraska and Zagotta, 2010*). Here, we validate our tmFRET approach using maltose-binding protein (MBP) as a model system. MBP is a bacterial periplasmic protein that undergoes a significant structural rearrangement upon binding maltose (*Figure 1A*; *Video 1*). The structure and rearrangement of MBP has been extremely well characterized, with over 260 structures in the protein structure database to date, both with and without ligands bound. Using both soluble and membrane-bound forms of MBP, we have determined the precision and accuracy of ACCuRET for measuring intramolecular distances and changes in distance due to ligand binding (*Figure 1*, *Video 1*). Using MBP we show that: (1)

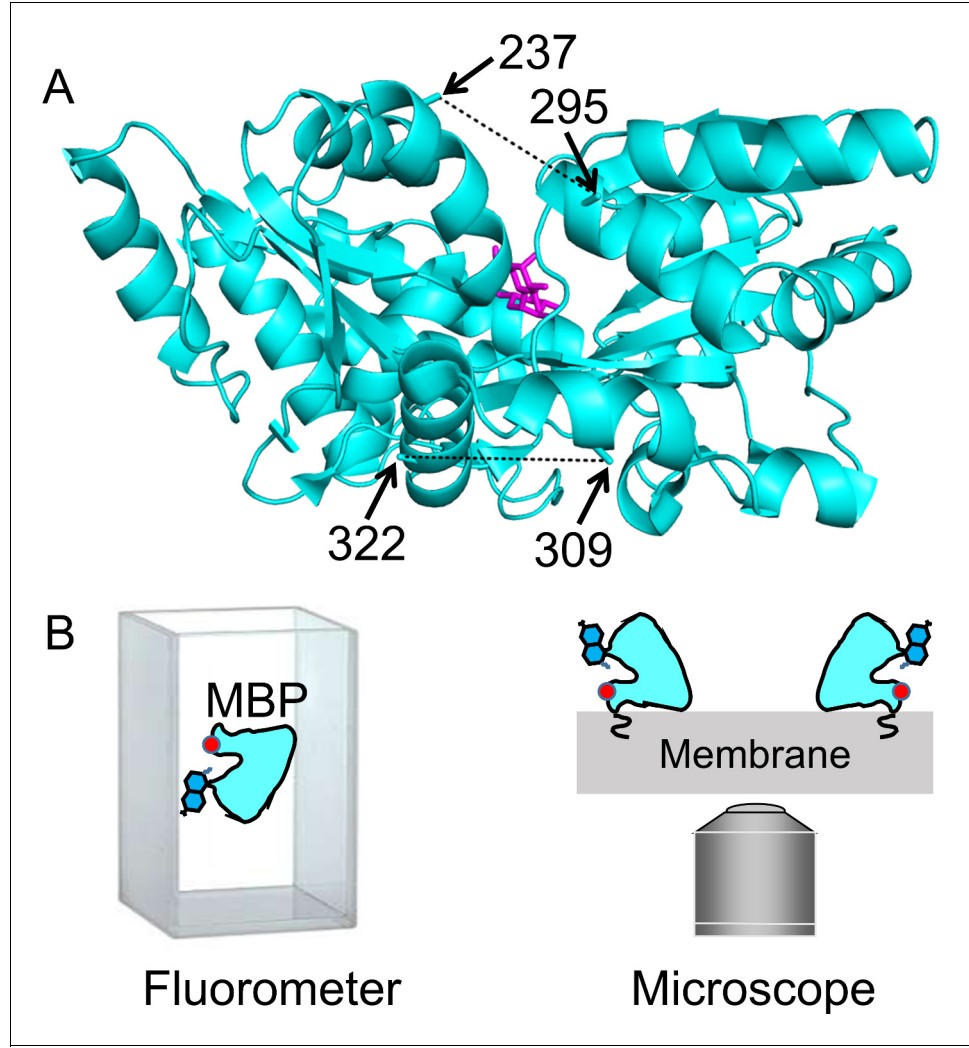

**Figure 1.** MBP as a model system for optimizing and validating ACCuRET. (**A**) X-ray crystal structure of MBP bound to maltose (PDB ID: 1N3X) showing the positions of the donor and acceptor FRET pairs at the top (295 and 237, respectively) and bottom (322 and 309, respectively) of the clamshell, with the distance represented by dashed lines. The distance separating the 295/237 FRET pair decreases and the 322/309 FRET pair increases with maltose binding (*Video 1*). (**B**) Soluble MBP was studied in a cuvette in a fluorometer (left), and membrane-bound MBP was studied in membranes of unroofed cells (right). The red dot represents bound metal, the blue side-chain represents L-Anap, and the squiggle at the membrane represents protein lipidation at the engineered CAAX site.
DOI: https://doi.org/10.7554/eLife.37248.002

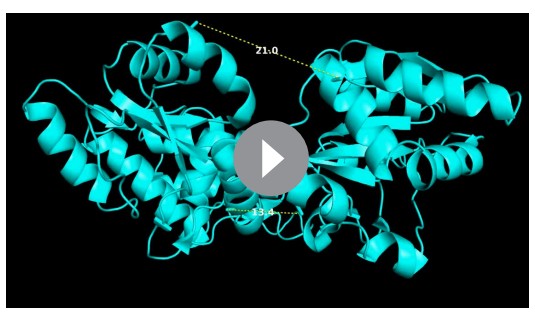

**Video 1.** Conformation change in MBP upon maltose binding. Morph of the X-ray crystal structures of MBP in the apo (PDB ID: 1N3W) and maltose-bound (PDB ID: 1N3X) states. Shown are the measurements of the distances for the FRET pairs used in this study, MBP-295Anap-C (top) and MBP-322Anap-C (bottom). Maltose is shown in magenta.
DOI: https://doi.org/10.7554/eLife.37248.003

ACCuRET provides an accurate measurement of intramolecular distances between 10 and 20 Å; (2) ACCuRET measures small changes in distance; (3) L-Anap can be introduced efficiently and specifically into proteins; (4) minimal metal binding sites can be introduced using a small, reversible cysteine reactive metal chelate; and (5) similar ACCuRET measurements can be made in membrane proteins in unroofed cells. These experiments establish this approach as a powerful method to measure the structural dynamics of membrane proteins.

## Results

### Selection of FRET pairs in MBP

MBP is a clamshell shaped protein that undergoes a significant closure of the clamshell upon binding its ligand, maltose (*Figure 1A*; *Video 1*). In addition, maltose binding leads to a smaller opening of the backside of the clamshell. By introducing one FRET pair at the top of the clamshell and a second FRET pair on the backside of the clamshell, we can address the accuracy and precision of ACCuRET to measure absolute distances as well as ligand-dependent increases and decreases in distance.

Our criteria for selection of FRET pairs were (1) the sites are solvent exposed, (2) the sites are on rigid secondary structural elements (α helices in MBP), (3) the distance between the sites is predicted to fall in the working range of tmFRET (~10–20 Å), and (4) the distance between sites undergoes a moderate change between apo and holo MBP. Our first FRET pair was at positions 295 (donor) and 237 (acceptor) on the outer lip of the clamshell (*Figure 1A*; *Video 1*). From β-carbon distances in the X-ray structures, these sites are 21 Å apart in the apo state and 13 Å apart in the holo state, a distance change of about −8 Å (shortening) with maltose binding. Our second FRET pair was at positions 322 (donor) and 309 (acceptor) on the backside of the clamshell (*Figure 1A*; *Video 1*). These sites are 13 Å apart in the apo state and 17 Å apart in the holo state, a distance change of +4 Å (lengthening) with maltose binding. In this study, we used these two FRET pairs to evaluate the accuracy and precision of ACCuRET with different acceptor metal-binding sites and different metals as well as for MBP in solution and bound to the membrane.

### Specific labeling with fluorophore

ACCuRET requires the protein to be site-specifically labeled with a donor fluorophore and an acceptor transition metal ion. To achieve specific labeling of MBP with a small donor fluorophore, we used amber codon suppression to introduce the fluorescent noncanonical amino acid L-Anap (*Figure 2A*) (*Chatterjee et al., 2013*; *Kalstrup and Blunck, 2013*). Our amber codon suppression method required the use of two plasmids (*Zagotta et al., 2016*; *Aman et al., 2016*). One plasmid encodes MBP with a TAG stop codon (the amber codon) engineered at the site for L-Anap incorporation. The second plasmid, pAnap, developed by Peter Schultz's lab (*Chatterjee et al., 2013*), encodes an evolved amino acyl tRNA synthetase and 4 copies of tRNA, which the cell uses to produce L-Anap-loaded tRNA complementary to the TAG codon (*Figure 2A*). HEK293T/17 cells were co-transfected with the two plasmids and incubated in a cell-permeable methyl ester (ME) version of L-Anap.

We achieved specific incorporation of L-Anap at donor positions 295 and 322 (*Figure 1A*). MBP-Anap was the primary band visible in gels of crude cell lysates using in-gel L-Anap fluorescence (*Figure 2—figure supplement 1*), indicating that L-Anap was not appreciably incorporated at other amber codon-containing sites. In addition, Western blots of FLAG-MBP containing the K295TAG or E322TAG mutation revealed full-length MBP only when the cells were incubated with L-Anap-ME (*Figure 2B,C*). However, even when L-Anap was present in the medium, a moderate amount of truncated product, in which translation terminated at the introduced stop codon, was observed for both

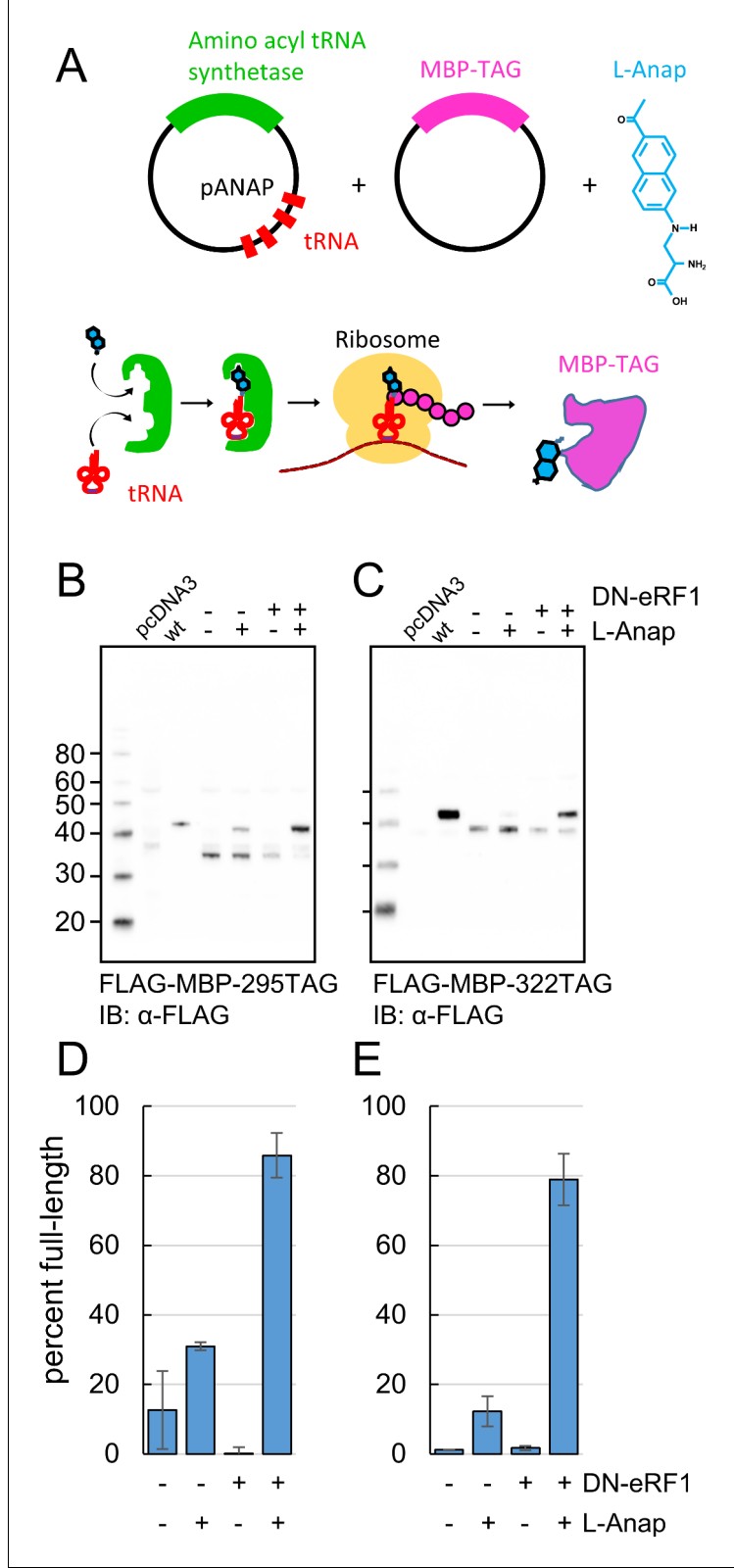

**Figure 2.** Incorporation of the fluorescent noncanonical amino acid L-Anap into MBP as the FRET donor. (**A**) Cartoon representation of amber codon suppression strategy for site-specific incorporation of L-ANAP into MBP. (**B–C**) Full-length protein for MBP-TAG constructs, where TAG indicates that the construct includes an internal amber stop codon, was produced with L-ANAP-ME and increased with DN-eRF1 cotransfection. Western blots are *Figure 2 continued on next page*

*Figure 2 continued*

shown of clarified lysates of cells cotransfected with (**B**) MBP-295TAG or (**C**) MBP-322TAG, pANAP, and either with (+) or without (-) DN-eRF1, and cultured with (+) or without (−) L-ANAP-ME in the medium. The blots were probed with an anti-FLAG antibody that recognizes both truncated and full length MBP. Vector only transfected cells (pcDNA3) were used as a negative control and wild-type MBP (wt) was used as a positive control. The amount of wild-type MBP loaded on the gel was less than the amount of the TAG mutations. As discussed in Materials and methods, all MBP constructs used in this work included an N-terminal FLAG epitope. (**D–E**) Quantitation of the percent of full-length product of (**D**) MBP-295TAG or (**E**) MBP-322TAG under the same conditions as B and C. Shown are mean ±SEM with n = 3.

DOI: https://doi.org/10.7554/eLife.37248.004

The following figure supplement is available for figure 2:

**Figure supplement 1.** SDS-PAGE showing purification of MBP constructs and specificity of Anap incorporation.
DOI: https://doi.org/10.7554/eLife.37248.005

MBP-295TAG and MBP-322TAG, as shown by the plots in *Figure 2D,E* (fraction of full-length MBP was 30% and 10%, respectively). These results indicate that L-Anap incorporation at positions 295 and 322 in MBP in HEK293T/17 cells was specific, but not particularly efficient.

To increase the efficiency of L-Anap incorporation, we co-expressed the MBP-TAG and pAnap plasmids with a third plasmid encoding a dominant negative form of eukaryotic Release Factor 1 (DN-eRF1). Previously, Jason Chin's group showed that DN-eRF1 increases the efficiency of incorporation of noncanonical amino acids in mammalian cells (*Schmied et al., 2014*), presumably by delaying release of the translation product from the ribosome when encountering an amber stop codon. For both sites in MBP, coexpression with DN-eRF1 substantially increased the fraction of full-length product from cells incubated with L-Anap-ME (from 30% to 86% for MBP-295TAG and from 10% to 79% for MBP-322TAG) (*Figure 2D,E*). Coexpression with DN-eRF1 also increased the absolute amount of full-length protein relative to that observed with wild-type MBP, from 1.6% (±0.4%; n = 3) to 6.3% (±0.5%; n = 3) for MBP-295TAG and from 5.5% (±4%; n = 3) to 50% (±27%; n = 3) for MBP-322TAG. We observed little or no read-through (i.e. incorporation of a natural amino acid at the stop codon), even when co-expressing DN-eRF1, as little full length MBP-TAG protein was observed when L-Anap was omitted from the cell growth medium (*Figure 2B,C*). These results establish a method for achieving specific, high-efficiency fluorescent labeling of MBP for ACCuRET experiments.

L-Anap's small size, short linker, and spectral properties makes it useful for tmFRET experiments (*Zagotta et al., 2016*; *Aman et al., 2016*; *Dai et al., 2018*; *Dai and Zagotta, 2017*). The peak absorption wavelength of L-Anap is 360 nm (*Chatterjee et al., 2013*). The emission of L-Anap is environmentally sensitive, blue shifting as the environment becomes less polar. The emission peak of L-Anap was 494 nm in our buffer and 475 nm in ethanol (see Materials and methods, Figure 13B). For the MBP-295Anap and MBP-322Anap in this study, the peak emissions were 491 nm and 478 nm, respectively (Figure 13A), suggesting that neither 295Anap nor 322Anap is fully solvent exposed and that the environment of 322Anap is more hydrophobic than that of 295Anap. We found that the quantum yield of L-Anap was also site dependent, measured to be 0.31 for MBP-295Anap and 0.47 for MBP-322Anap (see Materials and methods and Figure 13C). L-Anap's low quantum yield, together with its low extinction coefficient ($\sim 20 \times 10^3$ M$^{-1}$cm$^{-1}$; [*Chatterjee et al., 2013*; *Zagotta et al., 2016*]), makes L-Anap relatively dim compared to most visible light fluorophores; however, its fluorescence intensity is sufficient for most macroscopic fluorescence experiments. L-Anap's emission spectrum overlaps the absorption spectra of many transition metal ions (such as Ni$^{2+}$, Co$^{2+}$, and Cu$^{2+}$) making it a useful FRET donor for tmFRET (*Figure 3C*).

## Specific labeling with transition metal ion

The standard method to introduce a transition metal ion acceptor for tmFRET uses a dihistidine (HH) motif with histidines introduced at positions *i* and *i* + 4 on an α helix (*Figure 3A*) or at *i* and *i* + 2 on a βstrand (*Taraska et al., 2009a*; *Taraska et al., 2009b*; *Richmond et al., 2000*; *Puljung and Zagotta, 2013*; *Arnold and Haymore, 1991*; *Suh et al., 1991*). These minimal metal binding sites can bind Ni$^{2+}$, Co$^{2+}$, and Cu$^{2+}$ with affinities in the 1 to 100 μM range, and binding can be reversed with EDTA. Although this method places the transition metal ion very close to the protein backbone,

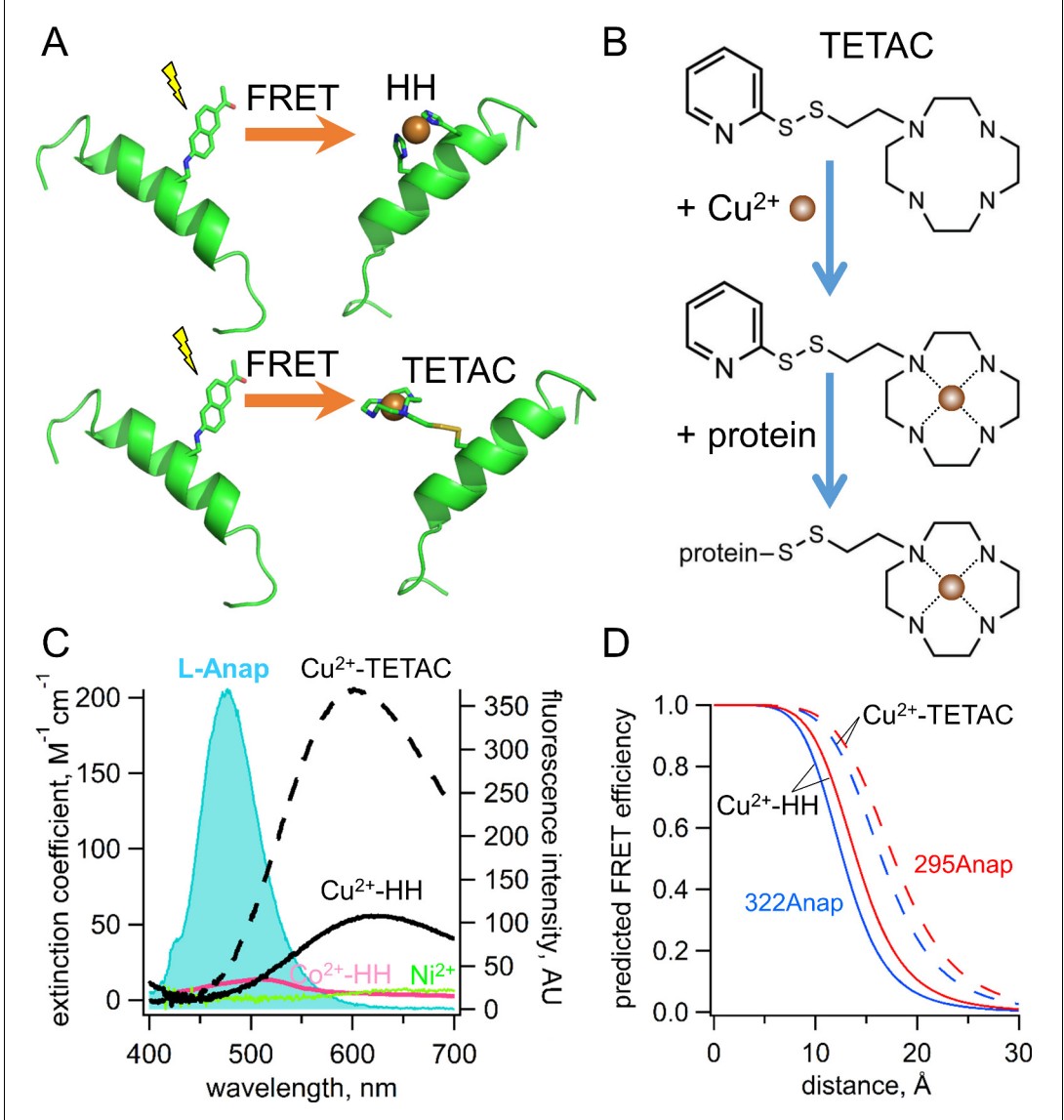

**Figure 3.** Incorporation of transition metal ions as the FRET acceptor using standard HH sites and TETAC. (**A**) Diagram illustrating tmFRET between L-Anap (left) and transition metal (right) for two different types of metal-binding sites: a standard HH site (top) and TETAC site (bottom). (**B**) Structure of TETAC, binding of $Cu^{2+}$ (brown ball), and reaction with a cysteine in a protein. (**C**) Spectral properties of free L-ANAP and $Cu^{2+}$-TETAC make them an ideal FRET pair for measuring small distances on the order of 10–20 Å. The emission spectrum from L-Anap (blue) is overlaid on the absorption spectra of $Cu^{2+}$-TETAC (dashed black line), $Cu^{2+}$-HH (black line), $Co^{2+}$-HH (magenta line) and $Ni^{2+}$ (green line). (**D**) Distance dependence of the FRET efficiency predicted with the Förster equation for each of the two FRET pairs, MBP-295Anap (red line) and MBP-322Anap (blue line), for both $Cu^{2+}$-TETAC (dashed line) and $Cu^{2+}$-HH (solid line).

DOI: https://doi.org/10.7554/eLife.37248.006

the metal concentrations required are high, so that they are not fully biocompatible and are subject to off-target binding.

To circumvent the limitations of standard HH sites, we developed a new, better method to introduce a specific, high-affinity metal-binding site in a protein by labeling a cysteine with 1-(2-pyridin-2-yldisulfanyl)ethyl)-1,4,7,10-tetraazacyclododecane (TETAC; *Figure 3A,B*). TETAC is a cysteine-reactive compound with a short linker to a cyclen ring that binds transition metal ions with very high-affinity (sub-nanomolar; [*Kodama and Kimura, 1977*]). Single cysteines can then be introduced in the protein at sites of interest and reacted with TETAC to create a mixed disulfide linkage to a cyclen metal-binding site.

ACCuRET uses TETAC to introduce minimal metal binding sites. TETAC has many advantages compared with standard HH sites for tmFRET: (1) it requires only nominal concentrations of free $Cu^{2+}$ eliminating off-target binding of $Cu^{2+}$ to endogenous binding sites on the protein; (2) it can be used with any type of secondary structural element; (3) its reaction with endogenous cysteines produces tmFRET only for sites that are very close (<20 Å), so endogenous cysteines need not be eliminated; (4) it can be easily removed with reducing agent, making its use compatible with physiological concentrations of other metals, for example $Ca^{2+}$; and (5) it significantly blue shifts and increases the absorbance of $Cu^{2+}$, extending the range of $R_0$ values available for tmFRET (*Figure 3C*). As shown in *Figure 3D*, the distance dependence of the FRET efficiency predicted by the Förster equation for each of our two L-Anap sites in MBP is shifter by almost 4 Å to longer distances for $Cu^{2+}$-TETAC compared to $Cu^{2+}$-HH. ACCuRET, therefore, offers a flexible and biocompatible method for measuring distances in the biological distance range.

For each of our two FRET pairs (*Figure 1A*), we made three constructs: one with no metal-binding site, one with a standard HH-binding site, and one with a Cys, for modification by TETAC. For MBP-295Anap, the HH was introduced at positions 233 and 237 (MBP-295Anap-HH) and the Cys was introduced at position 237 (MBP-295Anap-C). For MBP-322Anap, the HH was introduced at positions 305 and 309 (MBP-322Anap-HH) and the Cys was introduced at position 309 (MBP-322Anap-C). For both donor sites, control constructs were also made with no introduced metal-binding sites (MBP-295Anap and MBP-322Anap) to measure other forms of energy transfer and nonspecific decreases in fluorescence, and correct the FRET efficiency measurements accordingly (see Materials and methods). These constructs were all expressed in HEK293T/17 cells as amino-terminal FLAG fusions and purified using anti-FLAG beads for subsequent analysis (*Figure 2—figure supplement 1*).

## ACCuRET measures FRET and ligand-dependent changes in FRET

ACCuRET was readily observed in both MBP-Anap-C constructs. We recorded emission spectra before and after application of 10 µM $Cu^{2+}$-TETAC in a fluorometer (see Materials and methods and Approaches). As shown in *Figure 4*, $Cu^{2+}$-TETAC produced a large decrease in fluorescence intensity for both MBP-295Anap-C (*Figure 4A*) and MBP-322Anap-C (*Figure 4B*). The emission spectra in the presence of $Cu^{2+}$ were nearly identical in shape to the spectra without $Cu^{2+}$-TETAC (*Figure 4A–D*, dashed traces) indicating the fluorescence quenching reflected a FRET mechanism as opposed to a change of environment of L-Anap or an inner filter effect (*Lakowicz, 2006*).

Fluorescence quenching was strongly dependent on the presence of maltose. For MBP-295Anap-C, the quenching was greater with maltose indicating an increase in FRET efficiency and shorter distance (*Figure 4C*). For MBP-322Anap-C, the quenching was smaller with maltose indicating a decrease in FRET efficiency and longer distance (*Figure 4D*). Changes in tmFRET in MBP-295Anap-C therefore allowed us to visualize the closing of the top of the clamshell with ligand binding, and changes in tmFRET in MBP-322Anap-C allowed us to visualize the opening of the backside of the clamshell (*Video 1*). Importantly, maltose did not change the shape of the emission spectrum or shift the peak for either construct (*Figure 4E,F*), indicating the environment of L-Anap did not change upon maltose binding. However, because ACCuRET measures L-Anap fluorescence in the absence and presence of the acceptor for both the apo and ligand-bound states, it would account for any ligand-dependent changes in environment that would otherwise affect the calculation of FRET efficiency.

The maltose affinity of wild-type MBP is about 2 µM (*Martineau et al., 1990*). We found that MBP purified from HEK293T/17 cells with L-Anap incorporated at position 295 or 322, and $Cu^{2+}$ bound to HH or TETAC, appeared to be bound to an endogenous ligand, perhaps glycogen. These constructs exhibited FRET efficiencies expected for the maltose-bound states, and did not exhibit the maltose-dependent changes in tmFRET even after extensive washing and ion exchange chromatography (data not shown). Therefore, in all the MBP constructs in this paper we introduced a mutation into the maltose-binding site (W340A) that has previously been shown to reduce the maltose affinity (*Martineau et al., 1990*). This construct exhibited a maltose-dependent change in FRET (*Figure 4*) indicating that it did not copurify with an endogenous sugar.

We used ACCuRET to measure the maltose affinity of MBP-295Anap-C (containing W340A) modified with $Cu^{2+}$-TETAC by measuring the fluorescence quenching as a function of maltose concentration. As shown in *Figure 5*, maltose caused a concentration-dependent decrease in fluorescence with an apparent affinity of 280 µM, similar to the value measured previously with tryptophan fluorescence (*Martineau et al., 1990*). No decrease in fluorescence was observed in MBP-295Anap

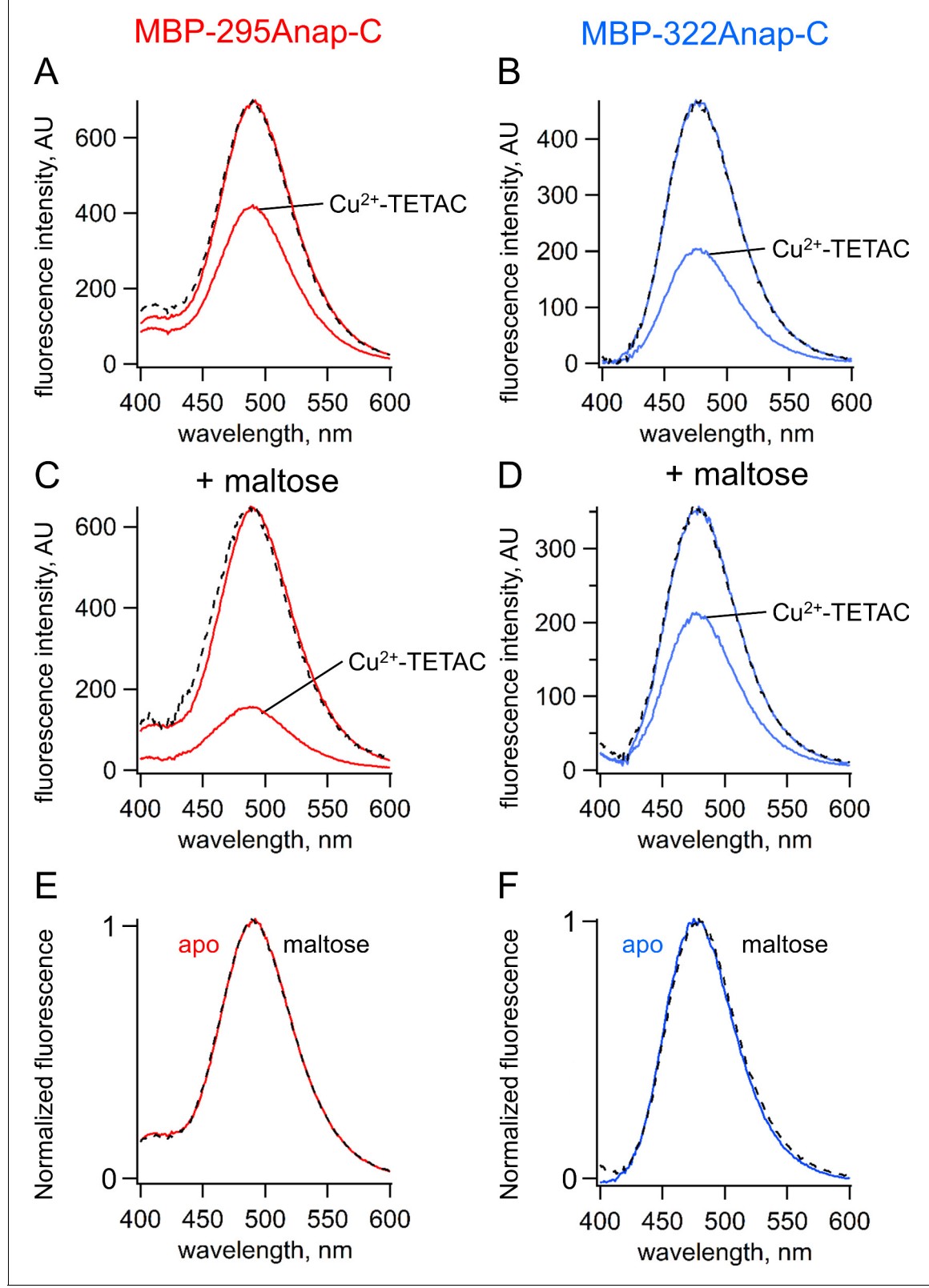

**Figure 4.** ACCuRET measures FRET and ligand-dependent changes in FRET. (**A–B**) Emission spectra of L-Anap for (**A**) MBP-295Anap-C and (**B**) MBP-322Anap-C measured in a fluorometer, before and after addition of 10 μM $Cu^{2+}$-TETAC to the cuvette. The fluorescence quenching reveals FRET between L-Anap and $Cu^{2+}$-TETAC. (**C–D**) Emission spectra of L-Anap for (**C**) MBP-295Anap-C and (**D**) MBP-322Anap-C, before and after addition of 10 μM $Cu^{2+}$-TETAC, in the presence of 10 mM maltose. Maltose produced an increased FRET in MBP-295Anap-C and decreased FRET in MBP-322Anap-

*Figure 4 continued on next page*

*Figure 4 continued*

C. The dashed lines in panels A-D represent the $Cu^{2+}$-TETAC spectra scaled to the initial spectra. The similarity indicates that quenching produced little or no change in the shape of the emission spectra, consistent with a FRET mechanism. (**E–F**) Scaled emission spectra of L-Anap for (**E**) MBP-295Anap-C and (**F**) MBP-322Anap-C with and without maltose. The similarity suggests that Anap experiences little or no environmental changes with maltose binding.

DOI: https://doi.org/10.7554/eLife.37248.007

lacking the introduced cysteine (***Figure 5***). These experiments illustrate the utility of ACCuRET for measuring both structural and functional properties of proteins.

To quantify ACCuRET, we measured the time course of the fluorescence at a given wavelength before and after addition of $Cu^{2+}$-TETAC and the reducing agent DTT. For both MBP-295Anap-C and MBP-322Anap-C, there was a rapid (<10 s) and substantial quenching of the fluorescence upon addition of 10 μM $Cu^{2+}$-TETAC (***Figure 6A,B***). The quenching was nearly completely reversed upon addition of DTT (***Figure 6A,B***) and negligible in the absence of the introduced cysteine (***Figure 6C, D***), indicating that it resulted from a reaction of $Cu^{2+}$-TETAC with the introduced cysteine. For MBP-295Anap-C the quenching was greater in the presence of maltose whereas for MBP-322Anap-C the quenching was less in the presence of maltose (***Figure 6C,D***), as expected from the predicted distance changes. These results establish that ACCuRET can serve as a useful alternative to tmFRET with standard HH sites to create a minimal metal-binding site for tmFRET with L-Anap and report intramolecular distance changes.

## tmFRET between L-Anap and transition metal bound to standard HH metal-binding sites

We compared tmFRET efficiencies measured with ACCuRET to those measured with $Cu^{2+}$ bound to standard HH sites, using the same donor-acceptor positions on MBP. Addition of 100 μM $Cu^{2+}$ caused a rapid quenching of both MBP-295Anap-HH and MBP-322Anap-HH fluorescence (***Figure 7A,B***). The quenching was nearly completely reversed upon addition of EDTA (***Figure 7A,B***) and negligible in the absence of the introduced HH sites (***Figure 7C, D***), indicating that it resulted from binding of $Cu^{2+}$ to the introduced HH sites. For MBP-295Anap-HH, the quenching was greater in the presence of maltose, whereas for MBP-322Anap-HH the quenching was less in the presence of maltose (***Figure 7***), as observed with $Cu^{2+}$-TETAC (***Figure 6***).

For both L-Anap sites, the quenching increased with increasing $Cu^{2+}$ concentration, following a simple binding isotherm, with an apparent affinity corresponding to the affinity of the metal-binding site (***Figure 7—figure supplement 1***). For $Cu^{2+}$ binding to MBP-295Anap-HH (in the presence of maltose), the affinity was 3 μM (***Figure 7—figure supplement 1C***), and for $Cu^{2+}$ binding to MBP-322Anap-HH (in the absence of maltose) the affinity was 17 μM (***Figure 7—figure supplement 1D***). The quenching at saturating concentrations reflects the FRET efficiency between L-Anap and the bound metal. These data support the use of 100 μM $Cu^{2+}$ as a

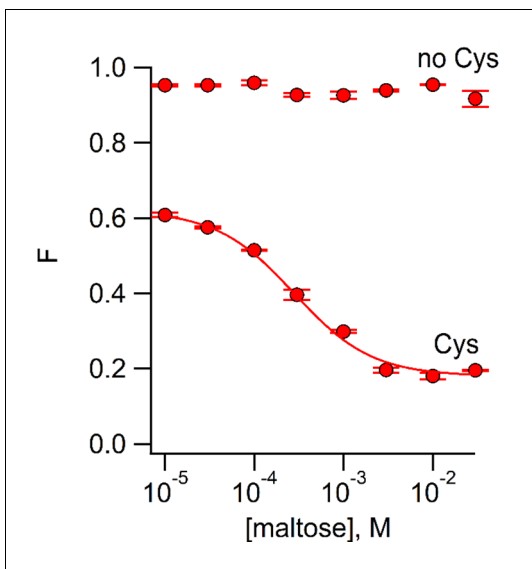

**Figure 5.** Using ACCuRET to measure the dependence of the rearrangement in MBP on maltose concentration. Plot of the fractional fluorescence of MBP-295Anap-C (Cys) and MBP-295Anap (no Cys), as indicated, after addition of $Cu^{2+}$-TETAC, as a function of maltose concentration. Shown are mean ±SEM for n = 4. The smooth curve is a fit of a binding isotherm with an apparent affinity of 280 μM. The low affinity is expected from the W340A mutation we introduced into all of our MBP constructs to reduce binding to endogenous sugars (***Martineau et al., 1990***).
DOI: https://doi.org/10.7554/eLife.37248.008

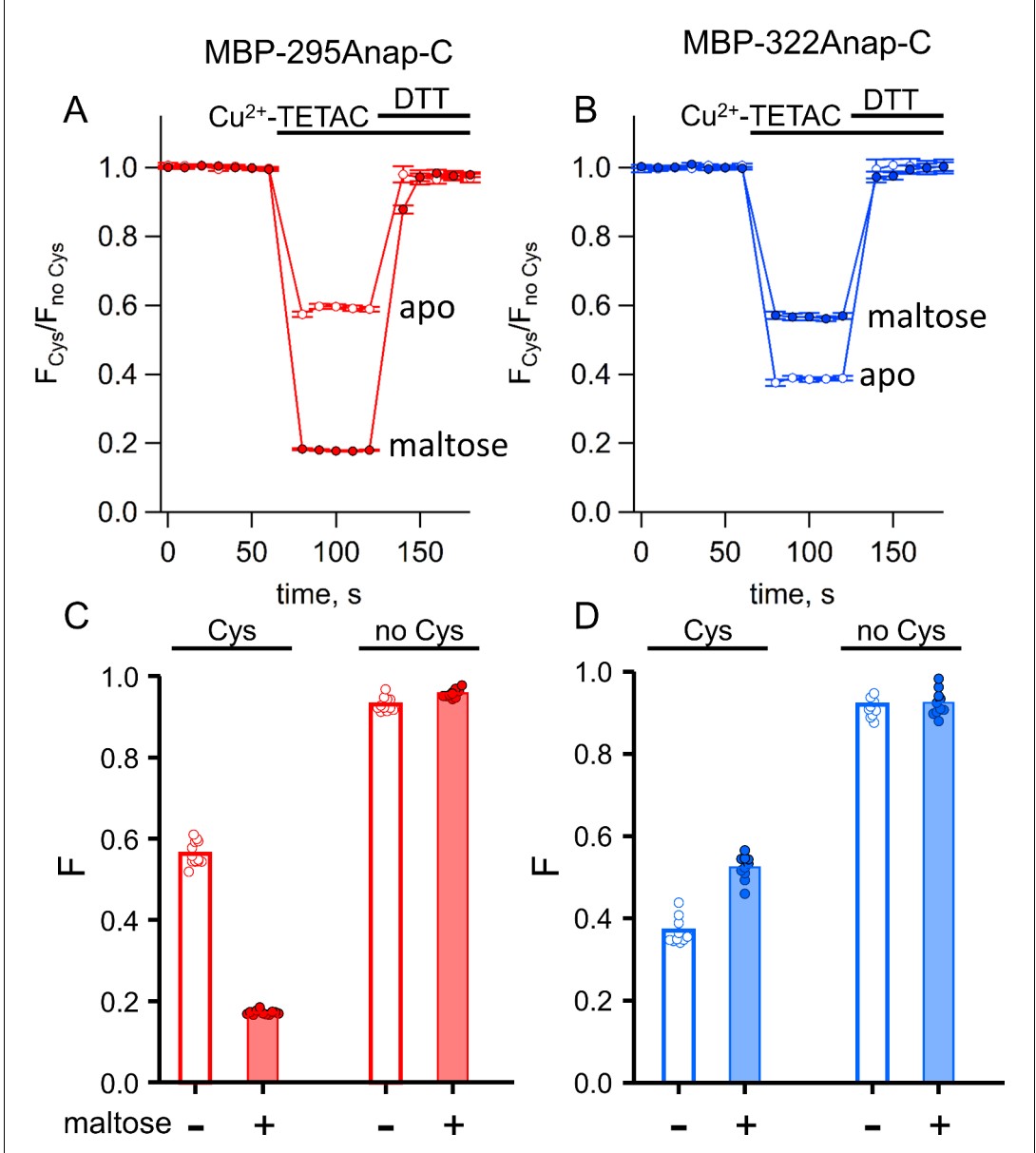

**Figure 6.** ACCuRET in MBP is reproducible, reversible, maltose dependent, and specific for each FRET pair. (A–B) Time course and reversal of ACCuRET for (A) MBP-295Anap-C and (B) MBP-322Anap-C in the absence (open symbols) and presence (filled symbols) of 10 mM maltose. The fractional fluorescence of each construct was recorded every 10 s and normalized to the fractional fluorescence of the corresponding construct without an introduced cysteine. 10 μM $Cu^{2+}$-TETAC and 1 mM DTT were added to the cuvette at the times indicated by the bars. Shown are mean ±SEM for n = 6–8. (C–D) Scattergrams of fractional fluorescence of (C) MBP-295Anap and (D) MBP-322Anap, with (Cys) and without (no Cys) the introduced cysteine, after addition of 10 μM $Cu^{2+}$-TETAC in the absence (-) and presence (+) of 10 mM maltose. The amount of quenching was reproducible, maltose-dependent, construct-dependent, and nearly absent without the introduced cysteine.

DOI: https://doi.org/10.7554/eLife.37248.009

saturating concentration for measuring tmFRET in these constructs.

Interestingly, at 1 mM $Cu^{2+}$ there was some fluorescence quenching in MBP-295Anap, but little in MBP-322Anap (**Figure 7—figure supplement 1**). As these constructs lack the HH metal-binding site and 1 mM $Cu^{2+}$ is a fairly high concentration, this quenching likely represents collisional quenching and suggests that 295Anap has a greater solvent accessibility compared to 322Anap, as also suggested by the emission spectra discussed above (Figure 13A). These data also indicate that there was negligible quenching by $Cu^{2+}$ bound to endogenous sites in MBP at $Cu^{2+}$ concentrations below

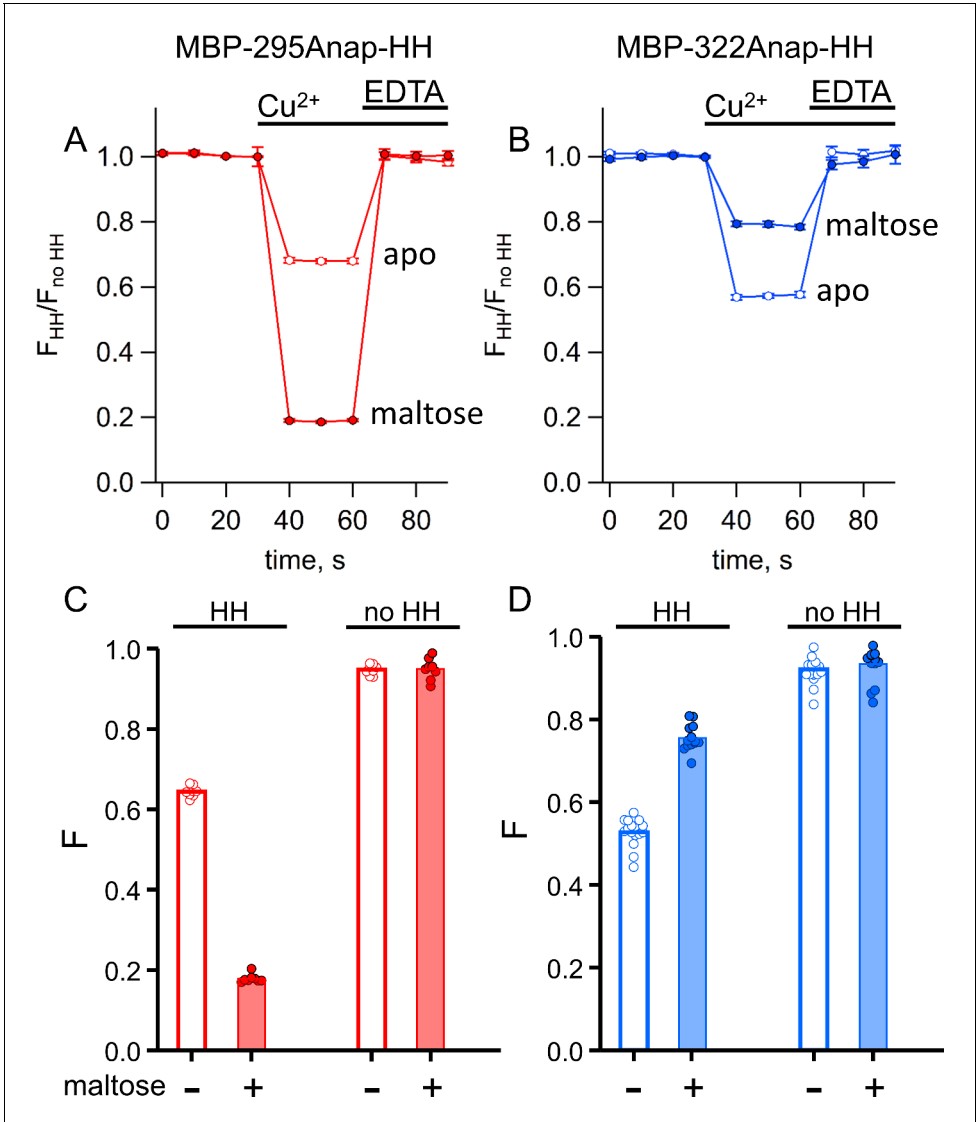

**Figure 7.** tmFRET with standard HH sites in MBP gives similar quenching to ACCuRET for the same FRET pairs. (A–B) Time course and reversal of tmFRET with HH sites for (A) MBP-295Anap-HH and (B) MBP-322Anap-HH in the absence (open symbols) and presence (filled symbols) of 10 mM maltose. The fractional fluorescence of each construct was recorded every 10 s and normalized to the fractional fluorescence of the corresponding construct without a HH site. 100 μM $Cu^{2+}$ and 10 mM EDTA were added to the cuvette at the times indicated by the bars. Shown are mean ±SEM for n = 8. (C–D) Scattergrams of fractional fluorescence of (C) MBP-295Anap and (D) MBP-322Anap, with (HH) and without (no HH) the HH site, after addition of 100 μM $Cu^{2+}$ in the absence (-) and presence (+) of 10 mM maltose. The amount of quenching with standard HH sites was similar to that with ACCuRET.

DOI: https://doi.org/10.7554/eLife.37248.010

The following figure supplement is available for figure 7:

**Figure supplement 1.** tmFRET with standard HH sites in MBP with different metals and metal concentrations.

DOI: https://doi.org/10.7554/eLife.37248.011

1 mM. Although not an issue with the constructs used here, it is worth noting that these potential problems caused by high concentrations of free $Cu^{2+}$ are largely mitigated in ACCuRET.

One advantage of standard HH sites is the ability to use transition metals with different coordination chemistries and absorption profiles than $Cu^{2+}$ (*Figure 3C*). These metals, including $Co^{2+}$ and $Ni^{2+}$, are expected to have different binding affinities for HH and endogenous sites as well as different $R_0$ values. $Co^{2+}$ produced a similar FRET efficiency to $Cu^{2+}$ but exhibited about a 10-fold lower binding affinity (*Figure 7—figure supplement 1C*, pink). $Ni^{2+}$, however, produced a substantially

lower FRET efficiency at saturating concentration (*Figure 7—figure supplement 1C*, green). This reflects the lower $R_0$ value predicted for $Ni^{2+}$ and further supports a FRET mechanism for the energy transfer. More importantly, the lower $R_0$ value for $Ni^{2+}$ expands the useable range of tmFRET measurements to shorter distances. With standard HH sites, metals with different $R_0$ values can be selected to match the distance of interest.

## Determination of distances and distance changes using ACCuRET

FRET efficiency is steeply dependent on distance between the donor and acceptor, making FRET a molecular ruler for measurements of distances (*Stryer and Haugland, 1967*). Accurate distance measurements, however, have historically been a challenge for FRET studies because they are highly dependent on the conformational heterogeneity and dynamics of the fluorophores, the relative orientation of the fluorophores, and the labeling specificity and efficiency (*Best et al., 2007*; *Schuler et al., 2005*; *Sobakinskaya et al., 2018*). ACCuRET was designed to address these problems by using small probes with short linkers, a metal ion acceptor that is isotropic, and labeling methods that are orthogonal, specific, and efficient.

Energy transfer from donor to acceptor causes quenching of donor fluorescence upon the addition of acceptor. With tmFRET, the FRET efficiency, E, can be easily calculated from the decrease in L-Anap fluorescence upon addition of metal acceptor. Other sources of energy transfer (e.g. solution quenching and off-target metal binding) and nonspecific decreases in fluorescence (e.g. bleaching and loss of protein) can confound calculations of FRET efficiency. The extent of this background quenching can be determined using constructs without an introduced metal-binding site and the FRET efficiencies corrected for as described in Materials and methods (*Taraska et al., 2009a*; *Dai and Zagotta, 2017*; *Puljung and Zagotta, 2013*).

From the FRET efficiencies, we calculated the distances between the donors and acceptors using the Förster equation $r = R_0(1/E - 1)^{1/6}$ where $R_0$ is the distance producing 50% FRET efficiency. We calculated $R_0$ using our measurements of the emission spectrum and quantum yield of L-Anap at each of the two positions in MBP, 295 and 322 (Figure 13), and the absorption spectrum of $Cu^{2+}$ bound to either a HH motif or cyclen (*Figure 3C*). We assume random orientations of the donor and acceptor ($\kappa^2 = 2/3$), a reasonable assumption when one member of the FRET pair is a metal ion (*Selvin, 2002*).

ACCuRET accurately measured intramolecular distances and distance changes, comparable to tmFRET using standard HH sites. *Figure 8* compares the tmFRET distance measurements for all our MBP constructs, with either $Cu^{2+}$-TETAC or $Cu^{2+}$-HH, in the absence (open circles) or presence (closed circles) of maltose. Also shown are the β-carbon distances predicted from the X-ray crystal structures of MBP in the absence and presence of ligand. Overall, the distances were highly accurate, with a root-mean-square deviation (RMSD) between measured distances and crystal structure distances of 1.5 Å for ACCuRET and 1.8 Å for tmFRET with standard HH sites. There was little variability in our measurements (*Figure 6C,D*, and *Figure 7C,D*), suggesting that the errors were more likely to be in our estimates of the probe distances from the β-carbon positions, or in assumptions of the Förster equation (see below), than in experimental variability due to the method. In addition, the changes in distances with maltose were similar in direction and magnitude to the changes in β-carbon distances predicted from the crystal structures. Finally, we did not detect a systematic difference between the distances measured with $Cu^{2+}$-HH and $Cu^{2+}$-TETAC (*Figure 8*). In conclusion, ACCuRET accurately determined the absolute distances and maltose-dependent distance changes for the FRET pair at the top of the clamshell, for which the distance decreased with maltose, and the FRET pair on the back of the clamshell, for which the distance increased with maltose.

## ACCuRET measurements of distances in membrane proteins

Membrane proteins present a major challenge for measuring protein structure and conformational dynamics. Compared to soluble proteins, membrane proteins typically express at much lower levels, are difficult to purify in their native state, and often require their native membrane environment to exhibit physiological structural and functional properties. ACCuRET is well suited to address these challenges because it leverages the exquisite sensitivity of fluorescence measurements, the efficient incorporation of L-Anap into proteins in mammalian cells, the ability to work on unpurified proteins,

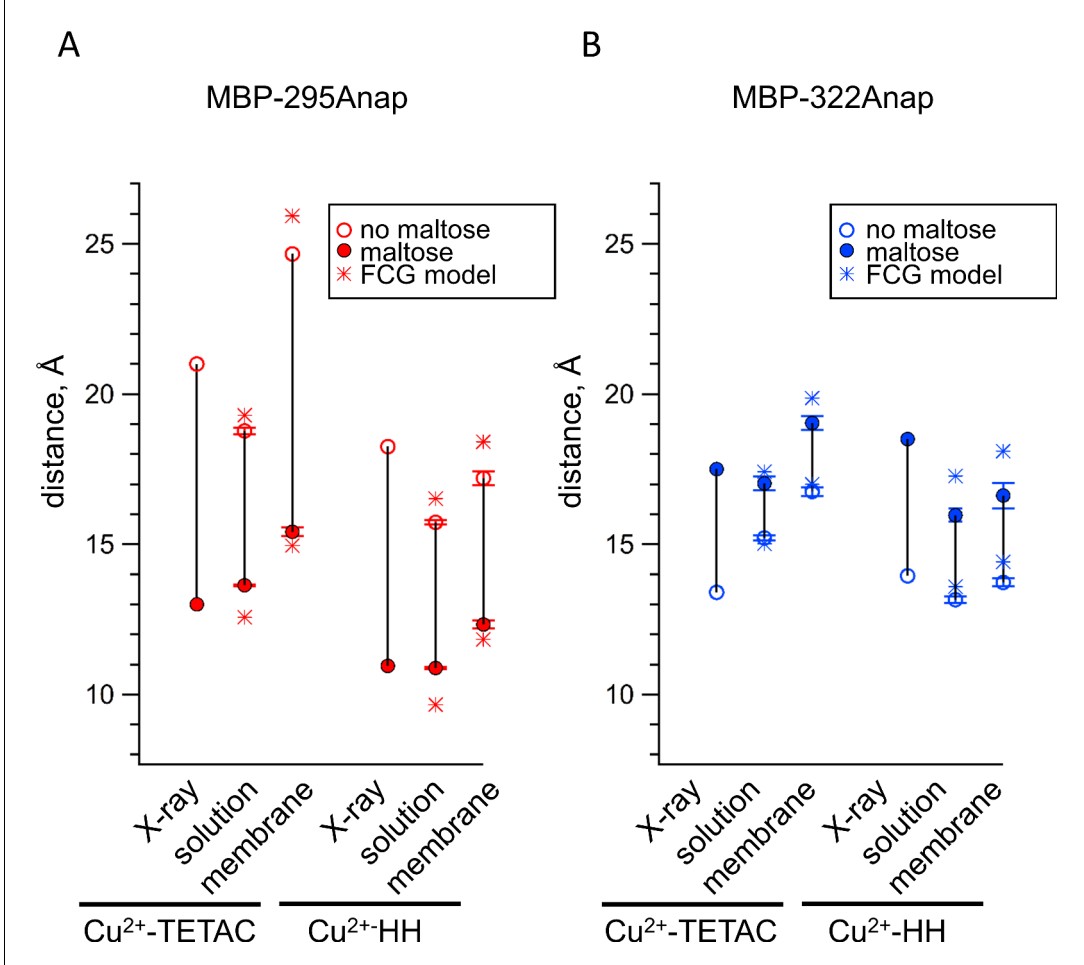

**Figure 8.** Comparison of distances measured with ACCuRET and standard HH sites to distances in the X-ray crystal structures of MBP. (A–B) Distances from the X-ray crystal structures (X-ray) and determined using ACCuRET ($Cu^{2+}$-TETAC) and standard HH sites ($Cu^{2+}$-HH) from soluble (solution) and membrane-bound (membrane) MBP for the (A) MBP-295Anap and (B) MBP-322Anap FRET pairs. The distances determined using the Förster equation in the absence (open symbols) and presence (filled symbols) of maltose are shown and connected by a vertical line that reflects the maltose-dependent change in distance. Shown are mean ±SEM for n $\geq$ 8. The distances determined using the FCG model, assuming FWHM = 8 Å, are shown as asterisks.
DOI: https://doi.org/10.7554/eLife.37248.012

and the ability to record from proteins in their native membranes. We therefore set out to validate and optimize ACCuRET for measuring conformational dynamics of proteins in the membrane.

To record fluorescence from native membranes we utilized a method called cell unroofing. Our implementation of this method utilizes a probe sonicator mounted on the stage of an inverted microscope to shear off the dorsal surface of cells and dislodge all soluble cellular contents and organelles (*Figure 9*; [*Zagotta et al., 2016*]). Cell unroofing leaves the ventral surface of cells intact as plasma membrane sheets attached to the coverslip for simple epifluorescence imaging. Cell unroofing also provides access to the intracellular surface of the membrane for application of transition metals and intracellular ligands. tmFRET measurements can then be made by recording the fluorescence from the unroofed cell before and after the application of $Cu^{2+}$-TETAC or free transition metal.

To validate and compare ACCuRET in unroofed cells, we made membrane-bound MBP constructs by adding a carboxy-terminal CAAX motif, which is then farnesylated in the cell (*Farnsworth et al., 1989*; *Anderegg et al., 1988*). These MBP-295TAG-CAAX and MBP-322TAG-CAAX constructs were each cotransfected into HEK293T/17 cells with pANAP and DN-eRF1 and incubated with L-Anap-ME in the medium, as described above. Both MBP-295Anap-CAAX and MBP-322Anap-CAAX expressed robustly and localized to the intracellular surface of the plasma membrane (*Figure 9—figure supplement 1*). Upon unroofing, the cells were no longer easily visible with bright field illumination.

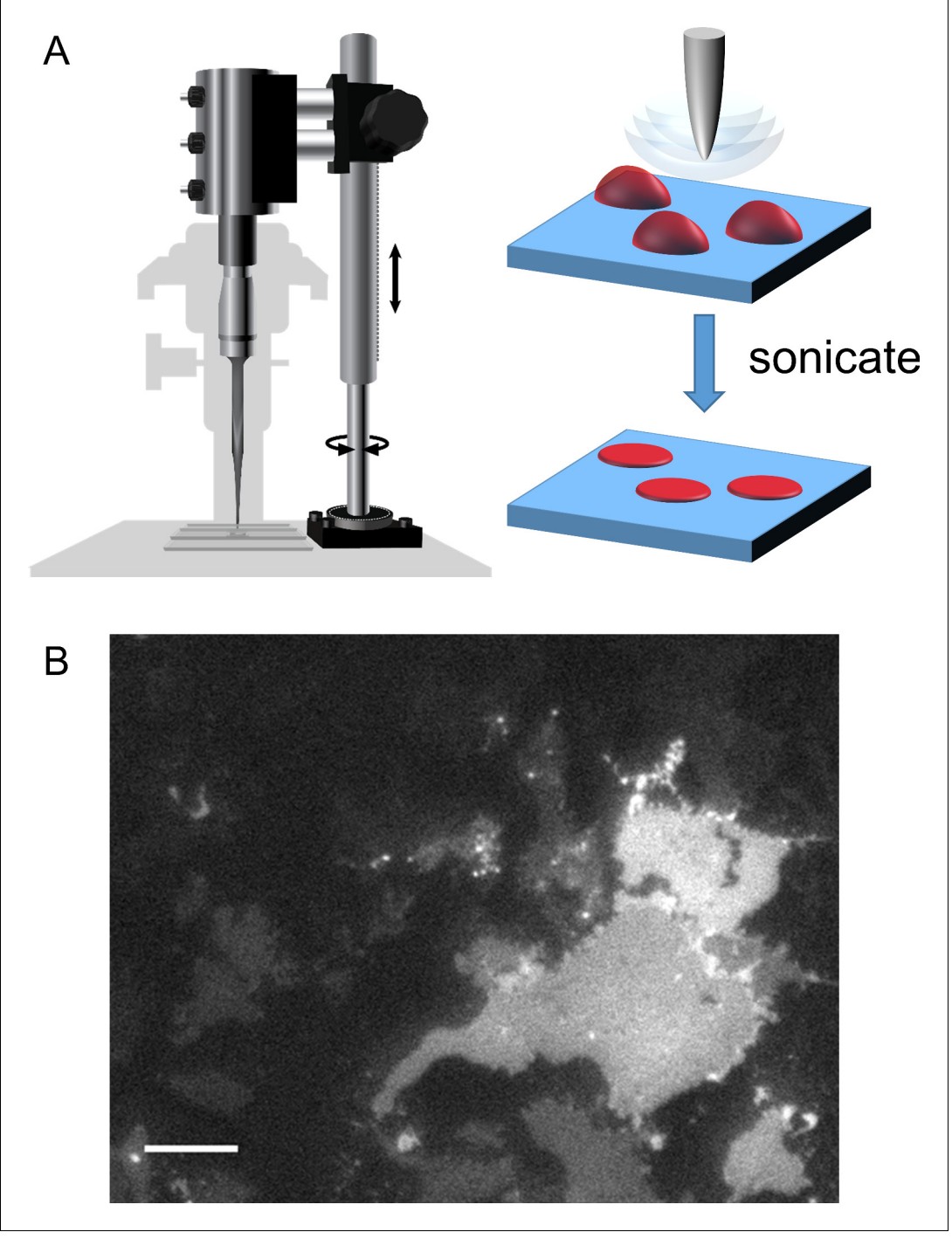

**Figure 9.** Cell unroofing to isolate plasma membrane sheets. (**A**) Diagram of the experimental setup used to unroof cells in a chamber on the microscope stage (left) with a cartoon of the unroofing process (right), as previously described (**Zagotta et al., 2016**). (**B**) An image of a field of unroofed HEK293T/17 cells transiently transfected with MBP-322TAG-CAAX and imaged with epifluorescence microscopy. The scale bar is 10 µm.
DOI: https://doi.org/10.7554/eLife.37248.013

The following figure supplement is available for figure 9:

**Figure supplement 1.** MBP-Anap-CAAX in intact cells.
DOI: https://doi.org/10.7554/eLife.37248.014

However, epifluorescence illumination showed plasma membrane sheets with relatively uniform L-Anap fluorescence (*Figure 9*) which were stable for >30 min (data not shown). Previously, we showed that these images look similar with total internal reflection fluorescence (TIRF) microscopy, confirming that most or all the fluorescence is from the plasma membrane (*Zagotta et al., 2016*). These results indicate that MBP-295Anap-CAAX and MBP-322Anap-CAAX proteins were stably associated with the membrane.

As for soluble MBP, $Cu^{2+}$-TETAC was an effective tmFRET acceptor for membrane-bound MBP-Anap in unroofed cells. We unroofed the MBP-Anap-C-CAAX constructs and recorded the L-Anap fluorescence before and after addition of $Cu^{2+}$-TETAC and DTT. As shown in *Figure 10A,C*, $Cu^{2+}$-TETAC produced a rapid decrease in L-Anap fluorescence of MBP-295Anap-C-CAAX that was reversed by the addition of DTT. The quenching was nearly absent in MBP-295Anap-CAAX without the introduced Cys (*Figure 10E*), indicating it arose from $Cu^{2+}$-TETAC modification of this Cys. The quenching was greater in the presence of maltose (*Figure 10C,E*) as expected from the decrease in distance between the donor and acceptor with maltose for the MBP-295Anap FRET pair (*Video 1*). Similarly, for MBP-322Anap-C-CAAX, $Cu^{2+}$-TETAC produced a rapid and reversible decrease in L-Anap fluorescence that was absent in MBP-322Anap-CAAX (*Figure 10B,D,F*). With MBP-322Anap-C-CAAX, however, the quenching was less in the presence of maltose (*Figure 10D,F*), as expected from the maltose-induced increase in distance between the donor and acceptor for the MBP-322Anap FRET pair (*Video 1*). The fractional fluorescence quenching was highly reproducible from cell to cell and was similar to the results obtained for soluble MBP in a cuvette (*Figure 6*). These results establish that ACCuRET can be used to measure tmFRET in membrane proteins.

We compared tmFRET efficiencies measured with ACCuRET to those measured with $Cu^{2+}$ bound to standard HH sites, using the same donor-acceptor positions. Addition of 100 µM $Cu^{2+}$ to unroofed cells expressing MBP-295Anap-HH-CAAX and MBP-322Anap-HH-CAAX produced a rapid decrease in fluorescence that was reversed by EDTA and nearly absent without the introduced HH sites (*Figure 11*). Furthermore, the quenching with $Cu^{2+}$ was greater with maltose than without maltose in MBP-295Anap-HH-CAAX and less with maltose than without maltose in MBP-322Anap-HH-CAAX. The fractional fluorescence quenching was highly reproducible from cell to cell and closely mirrored the results obtained for soluble MBP in a cuvette.

From the FRET efficiencies, we calculated the distances between the donors and acceptors using the Förster equation as described above. Comparing our determination of distances from membrane-bound MBP to those predicted from the X-ray crystal structures of MBP (*Figure 8*) gives RMSD values of 2.9 Å for ACCuRET and 1.3 Å for tmFRET with standard HH sites, comparable to the values for MBP in solution. However, distances calculated from membrane-bound MBP-Anap constructs were systematically longer than those determined from MBP-Anap constructs in solution (see Discussion). Together, our data demonstrate that ACCuRET provides accurate determinations of distances and distance changes in both soluble proteins and membrane proteins.

## Heterogeneity

The Förster equation predicts a steep distance dependence for FRET efficiency (*Figure 12A*, black curve) and assumes that the donor and acceptor probes are a fixed distance apart (*Figure 12B*, black). However, the linkers between the probes and the protein backbone produce substantial heterogeneity and mobility of the probes that effect the measured FRET efficiency (*Best et al., 2007*; *Schuler et al., 2005*; *Taraska et al., 2009b*). This heterogeneity in distance, together with the non-linear dependence of the FRET efficiency on distance, cause the distances closer to the $R_0$ distance to be weighted more heavily. The distance dependence of FRET efficiency, therefore, becomes shallower than predicted by the Förster equation. Our approach to reduce this effect has been to minimize these linkers by using a fluorescent amino acid and minimal metal binding sites closely associate with the backbone (HH or TETAC). Still, we observed a small, systematic bias in our tmFRET measurements where the distances greater than $R_0$ have somewhat greater FRET efficiency and the distances less than $R_0$ have somewhat lower FRET efficiency than predicted by the Förster equation (*Figure 12A*). This manifests as an apparent shallowing of the distance dependence and an underestimation of the changes in distance with maltose.

To correct for this heterogeneity, we developed a method we call FCG (<u>F</u>örster <u>c</u>onvolved with <u>G</u>aussian) analysis. FCG analysis assumes a Gaussian distribution of donor-acceptor distances in the sample (*Figure 12B*). This distribution is typical of the distance distributions between two spin labels

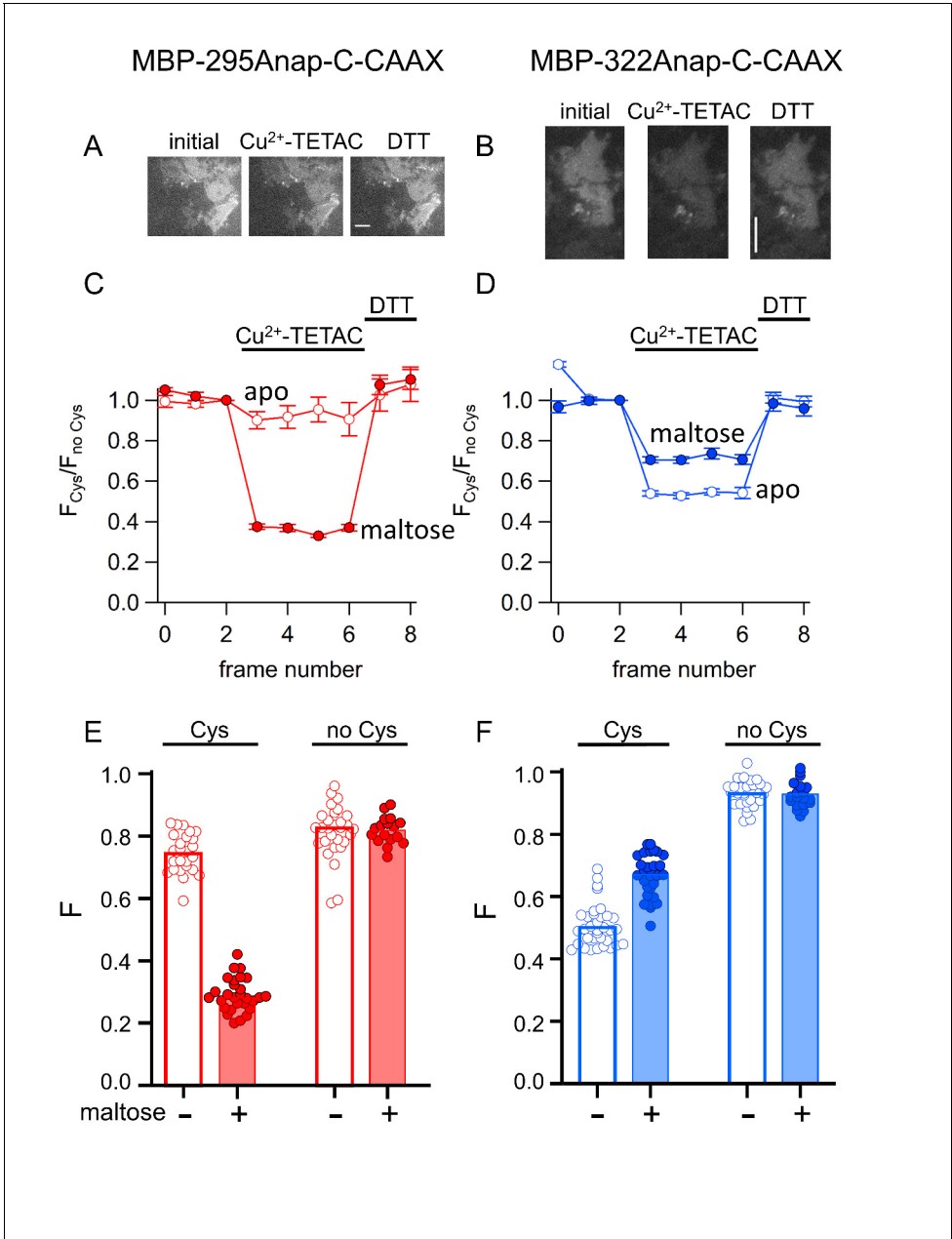

**Figure 10.** ACCuRET in membrane-bound MBP in unroofed cells. (**A–B**) Images of unroofed cells expressing (**A**) MBP-295Anap-C-CAAX or (**B**) MBP-322Anap-C-CAAX showing quenching upon addition of 10 μM $Cu^{2+}$-TETAC and reversal upon addition of 1 mM DTT, all in the absence of maltose. All images for a given field were acquired using the same illumination and camera settings and use the same look up table. Exposure time for (**A**) was 500 ms and for (**B**) was 125 ms. The scale bar is 10 μm. Time course and reversal of ACCuRET for (**C**) FLAG- MBP-295Anap-C-CAAX and (**D**) FLAG- MBP-322Anap-C-CAAX in the absence (open symbols) and presence (filled symbols) of 10 mM maltose. The fractional fluorescence of each construct was recorded every 10 s and normalized to the fractional fluorescence of the corresponding construct without an introduced cysteine. 10 μM $Cu^{2+}$-TETAC and 1 mM DTT were added to the chamber at the times indicated by the bars. The x-axis label, frame number, indicates the temporal sequence in which solutions were applied and images acquired. Shown are mean ±SEM for n ≥ 17. (**E–F**) Scattergrams of fractional fluorescence of (**E**) FLAG- MBP-295Anap-C-CAAX, and (**F**) FLAG- MBP-322Anap-C-CAAX, with (Cys) and without (no Cys) the introduced cysteine, after addition of 10 μM $Cu^{2+}$-TETAC in the absence (-) and presence (+) of 10 mM maltose. The amount of quenching was reproducible, maltose-dependent, construct dependent, and nearly absent without the introduced cysteine.

DOI: https://doi.org/10.7554/eLife.37248.015

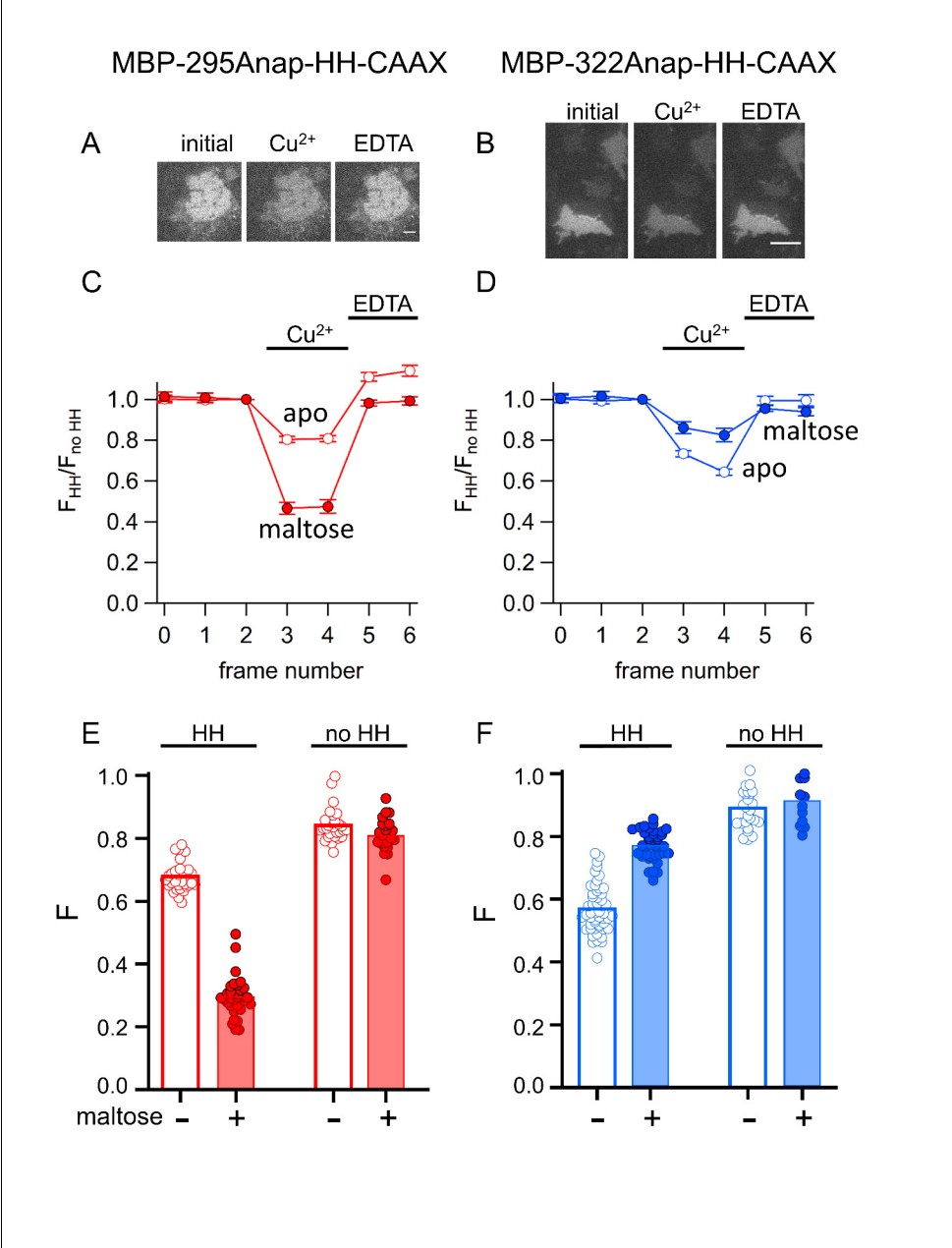

**Figure 11.** tmFRET with HH sites in membrane-bound MBP in unroofed cells. (**A–B**) Images of unroofed cells expressing (**A**) MBP-295Anap-HH-CAAX or (**B**) MBP-322Anap-HH-CAAX showing quenching upon addition of 100 µM $Cu^{2+}$ and reversal upon addition of 10 mM EDTA, all in the absence of maltose. All images for a given field were acquired using the same illumination and camera settings and use the same look up table. Exposure times for (**A**) and (**B**) were 125 ms. The scale bar is 10 µm. (**C–D**) Time course and reversal of tmFRET with HH sites for (**C**) MBP-295Anap-HH-CAAX or (**D**) MBP-322Anap-HH-CAAX in the absence (open symbols) and presence (filled symbols) of 10 mM maltose. The fractional fluorescence of each construct was recorded and normalized to the fractional fluorescence of the corresponding construct without a HH site. 100 µM $Cu^{2+}$ and 10 mM EDTA were added to the chamber at the times indicated by the bars. The x-axis label, frame number, indicates the temporal sequence in which solutions were applied and images acquired. Shown are mean ±SEM for n ≥ 12. (**E–F**) Scattergrams of fractional fluorescence of (**E**) MBP-295Anap-HH-CAAX or (**F**) MBP-322Anap-HH-CAAX, with (HH) and without (no HH) the HH site, after addition of 100 µM $Cu^{2+}$ in the absence (-) and presence (+) of 10 mM maltose. The amount of quenching with standard HH sites was similar to that with ACCuRET.

DOI: https://doi.org/10.7554/eLife.37248.016

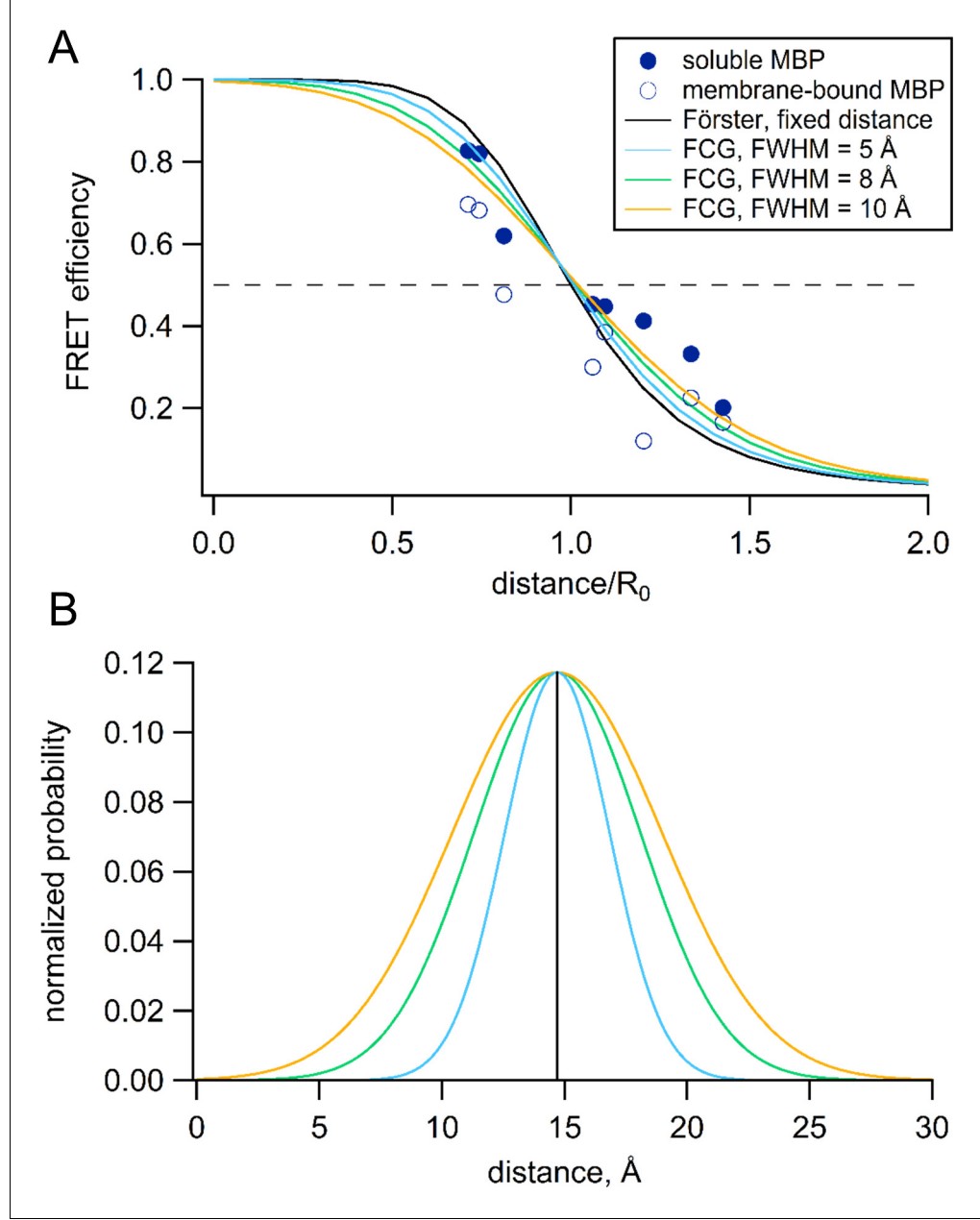

**Figure 12.** Förster equation convolved with Gaussian distribution to predict average distances. (**A**) Plot of the measured and predicted distance dependence of FRET efficiency. The FRET efficiencies measure in this paper from soluble (filled symbols) and membrane-bound (open symbols) MBP are plotted versus the Cβ-Cβ distances predicted from the MBP X-ray crystal structures (PDB IDs: 1N3W and 1N3X for apo and maltose-bound, respectively). The predicted distance dependencies are shown for the Förster equation and the Förster equation convolved with Gaussian distribution. The Gaussians used for convolution had mean intramolecular distances of 14.7 Å and the following FWHM values: 0 Å (black), 5 Å (blue), 8 Å (green), and 10 Å (orange). (**B**) The Gaussian distributions that were convolved with the Förster equation for panel (**A**), using the same color scheme.
DOI: https://doi.org/10.7554/eLife.37248.017

on rigid proteins measured using double electron-electron resonance (DEER) spectroscopy (*Jeschke, 2012*). When the Förster equation was convolved with a Gaussian distribution, the apparent FRET efficiency has a similar (but not identical) $R_0$ but a more shallow distance dependence than the Förster equation (*Figure 12A*). The greater the width of the Gaussian distribution, the shallower the distance dependence. Using a Gaussian distribution with a full width at half maximum (FWHM)

of 8 Å, we found that, in most cases, the distances and changes in distance calculated from the FCG model (*Figure 8*, asterisks) more closely matched the distances predicted from the crystal structures than the distances calculated with the Förster equation alone. Since all proteins at physiological temperature possess some structural heterogeneity (*Frauenfelder et al., 1991*), we propose that this correction may be useful for many FRET studies of distance.

## Discussion

In this paper, we introduced a new method, called ACCuRET, for measuring intramolecular distances and dynamics in proteins using tmFRET between a donor fluorophore and an acceptor transition metal ion. For the donor fluorophore, ACCuRET utilizes a fluorescent noncanonical amino acid, L-Anap, site-specifically incorporated into the protein using amber codon suppression. Orthogonal labeling with metal acceptor was achieved by modifying an introduced cysteine with $Cu^{2+}$-TETAC. Using MBP as a benchmark, we show that ACCuRET accurately measures absolute distances and changes in distance that faithfully represent the backbone dynamics underlying function in soluble and membrane proteins. Measurements of distance display an RMSD of 1.5–2.9 Å between our experimentally determine distances and the X-ray crystal structures. These experiments validate ACCuRET as a powerful new method for measuring structural dynamics of proteins in their native environment.

The noncanonical amino acid L-Anap has been previously used to label membrane proteins. However, most of these studies focused on state-dependent changes in fluorescence due to the environmentally-sensitive emission of L-Anap (*Kalstrup and Blunck, 2013*; *Dai et al., 2018*; *Dai and Zagotta, 2017*; *Soh et al., 2017*; *Sakata et al., 2016*; *Wen et al., 2015*; *Wulf and Pless, 2018*). L-Anap has also been recently used as a FRET donor in studies to discern movement in ion channels between domains or between the channel and the membrane (*Zagotta et al., 2016*; *Aman et al., 2016*; *Dai et al., 2018*; *Dai and Zagotta, 2017*). Although these studies demonstrated the potential power of using L-Anap as a donor for tmFRET, the distances and distance changes measured were difficult to interpret, given the limited structures available. By using MBP as a benchmark, we have been able to assess the accuracy of tmFRET for measurements of distances and changes in distance.

The use of amber codon suppression to introduce noncanonical amino acids has become more common in prokaryotic expression systems, but has been limited in eukaryotic expression systems by the much poorer efficiency of incorporation observed. We found that cotransfection of a plasmid encoding our TAG-containing MBP with one encoding the DN-eRF1 was critical in achieving sufficiently high levels of expression (*Schmied et al., 2014*). In addition to increasing the total amount of protein, DN-eRF1 also enhanced the fraction of the protein that was full-length without significant incorporation of natural amino acids at the amber codon site. Furthermore, in-gel fluorescence showed that, when L-Anap is present in the medium, MBP-Anap is the only major protein species observed. Thus, DN-eRF1 extends the power of amber codon suppression to mammalian cells and is particularly useful for eukaryotic membrane proteins that typically cannot be expressed in bacteria.

ACCuRET uses a novel metal-labeling strategy for tmFRET that is complementary, and in some ways superior to, the standard method of labeling via HH sites. $Cu^{2+}$-TETAC has previously been used in paramagnetic relaxation enhancement (PRE) and DEER experiments (*Cunningham et al., 2015*; *Sengupta et al., 2015*). The DEER experiments showed that it has a narrow inter-label distance distribution, very similar to that of the widely used spin label MSL and much narrower than MTS-EDTA (*Cunningham et al., 2015*). TETAC labels cysteines in any secondary structure and has a subnanomolar affinity for $Cu^{2+}$ (*Kodama and Kimura, 1977*), allowing ACCuRET to be implemented in a wider variety of conditions. Native cysteines are mostly compatible with the use of TETAC. Only native cysteines within ~20 Å of the L-Anap, or for which modification produces functional effects, are problematic. Such native cysteines can easily be identified using the control construct without the introduced cysteine, and only those that are problematic need be mutated. Because the affinity of $Cu^{2+}$ for TETAC is so high, $Cu^{2+}$-TETAC can be rinsed from the bath solution prior to imaging unroofed cells, allowing measurements to be made in the absence of free $Cu^{2+}$. Labeling with $Cu^{2+}$-TETAC can be reversed with a reducing agent such as DTT, obviating the need for a metal chelator such as EDTA that can strip off endogenous metals and divalent ions. Although we expected the position of $Cu^{2+}$ to be less constrained when bound to TETAC compared to HH, our distance measurements were as accurate for $Cu^{2+}$-TETAC as for $Cu^{2+}$-HH (*Figure 8*). Finally, binding to the cyclen

ring of TETAC increases the absorption of $Cu^{2+}$ and blue shifts the peak of its spectrum relative to $Cu^{2+}$-HH by 20 nm (*Figure 3C*), giving $Cu^{2+}$-TETAC an $R_0$ value almost 4 Å longer than $Cu^{2+}$ bound to HH (*Figure 3D*). This longer $R_0$ value increases the range of distance measurements that can be accurately measured with tmFRET.

Previously, the same donor/acceptor sites on MBP we use here have been studied with tmFRET using standard methods of labeling (*Yu et al., 2013*). At the donor site, the fluorophore fluorescein-5-maleimide or monobromobimane was reacted with an introduced cysteine, and, at the acceptor site, $Cu^{2+}$ or $Ni^{2+}$ was bound to a HH site. For the same donor/acceptor sites, our measurements of distances and changes in distance with ACCuRET were comparable with the measurements made using bimane as the donor, but more accurate than the measurements made using fluorescein as the donor, emphasizing the need for small donor fluorophores with short linkers. Importantly, however, these previous experiments required the use of purified protein with no native cysteines or metal-binding sites. By employing a fluorescent noncanonical amino acid as the donor, ACCuRET can be performed on unpurified protein in its native environment with most or all of the native cysteines and weak metal-binding sites left intact.

The dearth of approaches for studying conformational dynamics of proteins in their native environment makes ACCuRET especially valuable for studies of membrane proteins such as ion channels and transporters. We therefore compared ACCuRET in membrane-bound MBP in unroofed cells to soluble, affinity-purified MBP. Overall, the distance measurements in the membrane-bound MBP were similar to those in soluble MBP (*Figure 8*). We observed, however, a trend toward slightly lower FRET efficiency for membrane-bound MBP than observed with soluble MBP (*Figure 12A*). The most likely explanation for this difference, which translated into ~2–3 Å increase in distance for our measurements in membrane-bound MBP compared to soluble MBP, is that our background subtraction method (see Materials and methods) did not fully account for all the L-Anap-dependent background signal. One potential source of L-Anap-dependent background that would lead us to underestimate FRET efficiency is incomplete labeling with the acceptor transition metal. This could occur if, for example, oxidation or post-translational modification of the introduced cysteine prevented it from reacting with TETAC. A number of our results indicate this was not, however, a concern. We observed very similar distances with $Cu^{2+}$ bound to HH sites and $Cu^{2+}$ bound to TETAC, indicating that the $Cu^{2+}$-TETAC reaction with cysteines was as efficient as $Cu^{2+}$ binding to HH sites. We also measured the reaction kinetics of $Cu^{2+}$-TETAC at each site to ensure the reaction had gone to completion. Finally, $Cu^{2+}$-TETAC quenched >80% of L-Anap fluorescence in MBP-295Anap-C in the presence of maltose, and the remaining fluoresacence signal is what is expected from the known donor-acceptor distance and the $R_0$ of the FRET pair. Determining the remaining source of background fluorescence and developing better corrections will further improve the accuracy of ACCuRET for membrane proteins and, perhaps, proteins in intact cells.

ACCuRET mitigates a number of limitations that have hampered measurements of FRET efficiencies and calculations of absolute distances. Because ACCuRET measures L-Anap fluorescence in the absence and presence of the acceptor for both the apo and ligand-bound states, it accounts for any ligand-dependent changes in environment that would otherwise affect the calculation of FRET efficiency. Our measurements of the emission spectra of L-Anap incorporated at the two sites in MBP allowed us to use the $R_0$ determined for each site in our calculations of distance. The difference in $R_0$ between MBP-295Anap (17.7 Å) and MBP-322Anap (16.5 Å), however, was small and would not have substantially affected our estimates of distances or changes in distance. The difference in quantum yield between our two sites (MBP-295Anap:0.31 and MBP-322Anap:0.47) is larger than generally observed for the state-dependent changes in fluorescence at any given site, suggesting that state-dependent changes in $R_0$ are not usually a concern.

Experimental limitations that could compromise the accuracy of distance determinations include: a broad distance distribution between donor and acceptor; measuring distances (*r*) outside the range of $\sim 1.5 R_0 > r > \frac{R_0}{1.5}$ (*Figure 12A*); nonspecific labeling with the donor; incomplete labeling with the acceptor; and unexpected sources of background fluorescence. ACCuRET mostly overcomes these limitations. Using small probes with short linkers narrows the distance distribution, and FCG analysis improves distance determinations for distributed distances. The increase in absorption of $Cu^{2+}$-TETAC, and the corresponding increase in $R_0$ values, expands the utility of tmFRET over a broader distance range. Using amber codon suppression to introduce the donor substantially

reduces nonspecific labeling with donor compared to using cysteine-reactive donors for labeling. Using a slight excess of $Cu^{2+}$ with TETAC minimizes incomplete labeling with the acceptor. As discussed above, unexpected sources of background fluorescence appear to reduce the accuracy of distance measured for membrane-bound MBP, but this effect was small in our experiments.

The distance dependence of FRET is, in practice, less steep than predicted by the Förster equation (*Figure 12A*). This is at least partly explained by the idealized assumption of the Förster equation that relative distances between donors and acceptors are homogeneous (*Best et al., 2007*; *Schuler et al., 2005*). In fact, proteins have been shown to exhibit significant heterogeneity (*Frauenfelder et al., 1991*), with distances between side-chains well described by normal distributions (*Jeschke, 2012*). Convolving the Förster equation with Gaussian distance distributions (FCS analysis) gives distance-dependence curves that are less steep than the Förster equation itself (*Figure 12A*). From DEER studies, the distribution of distances between $Cu^{2+}$ ions bound to TETAC were well described by Gaussian distributions with FWHM values ranging from 6 to 9 Å (*Cunningham et al., 2015*). We used FCG analysis to convert FRET efficiencies to distances. These distances (*Figure 8*, asterisks calculated using FWHM = 8 Å) more closely match the donor-acceptor distances, as well as the maltose-induced distance changes, determined from the Cβ-Cβ values from X-ray crystal structures. Although we did not measure the distributions of donor-acceptor distance in our experiments, it seems clear that assuming a distribution of distances in the range found in the literature is a better assumption than assuming a fixed distance.

In summary, these experiments establish a new method called ACCuRET for measuring structural dynamics of proteins in their native environment, particularly membrane proteins. The method can measure distances with an accuracy of 1.5–2.9 Å and has the potential to measure structural dynamics on a time scale of milliseconds. For ion channels and transporters, ACCuRET can also be combined with patch-clamp fluorometry (PCF) to measure protein structure and function simultaneously. Although we used the unnatural amino acid L-Anap, our approach could employ fluorophores introduced with other unnatural amino acids (perhaps called unACCuRET). Ultimately, better fluorophores will enable tmFRET measurements with faster time resolution and single-molecule sensitivity.

## Materials and methods

### Key resources table

| Reagent type (species) or resource | Designation | Source or reference | Identifiers | Additional information |
|---|---|---|---|---|
| Cell line (*Homo sapiens*) | HEK293T/17 | ATCC | ATCC: CRL-11268; RRID:CVCL_1926 | |
| Recombinant DNA reagent | pANAP | Addgene: DOI: 10.1021/ja4059553 | Addgene: 48696; | |
| Recombinant DNA reagent | DN-eRF1 (peRF1-E55D.pcDNA5-FRT) | Jason Chin: DOI: 10.1021/ja5069728 | | |

### Constructs, cell culture, and transfection

All constructs were made in the pcDNA3.1 mammalian expression vector (Invitrogen, Carlsbad, CA), except as noted below. The amino acid sequence of MBP was based on the pMAL-c5x vector (New England Biolabs, Ipswich, MA), and was codon optimized for mammalian cell expression. To reduce the affinity of MBP for endogenous ligands and allow subsequent testing of maltose binding, we introduced a point mutation, W340A (*Martineau et al., 1990*), in all constructs. In addition, a FLAG epitope was added to the N-terminus of MBP. FLAG-MBP cDNA was synthesized by Bio Basic (Amherst, NY). The FLAG epitope is present in all constructs but is omitted from the construct names except in the discussion of western blot analysis. For membrane localization, the C-terminus of FLAG-MBP was fused to a CAAX domain with the following sequence: KMSKDGKKKKKKSKTKCVIM. The pAnap vector ([*Chatterjee et al., 2013*]; purchased from Addgene, Cambridge, MA) was used as previously described (*Zagotta et al., 2016*). The dominant negative eukaryotic release factor one construct (DN-eRF1) was kindly provided by Jason Chin (Cambridge, UK) and used in the provided pcDNATM5/FRT/TO vector (*Schmied et al., 2014*).

Amber stop codons (TAG), histidines and cysteines were introduced using standard PCR and oligonucleotide-based mutagenesis. All sequences were confirmed using automated DNA sequencing (Eurofin/Operon, Louisville, KY). The stop codons were introduced into FLAG-MGP at positions 295 (MBP-295TAG) and 322 (MBP-322TAG). For each stop codon, three constructs were produced: no metal-binding site, a di-histidine, and a cysteine. For MBP-295TAG, the di-histidines were introduced at positions 233 and 237 (MBP-295TAG-HH), and the cysteine was introduced at position 237 (MBP-295TAG-C). MBP-295TAG-C also contained the 233H mutation. For MBP-322TAG, the di-histidines were introduced at positions 305 and 309 and the cysteine at position 309.

HEK293T/17 cells were obtained from ATCC (Manassas, VA; #CRL-11268; RRID: CVCL_1926), and were expanded only once immediately upon receipt. Cells were used for no more than 35 passages, and replenished from frozen aliquots of the originally expanded stock. Aliquots of the originally expanded stock were used for no more than 8 years, at which point a new stock was purchased. No additional authentication was performed. Testing for mycoplasma contamination was performed using the MicroFluor Mycoplasma Detection kit (catalog #M7006; ThermoFisher Scientific, Waltham, MA) at the end of the study and cells were found to be contaminated.

HEK293T/17 cells were plated in 6-well trays on glass coverslips treated with poly-lysine. Cells were transfected at ≈25% confluency with a total of 1.6 µg of DNA and 10 µL of Lipofectamine 2000 (Invitrogen, Carlsbad, CA) per well. The 1.6 µg of DNA consisted of 0.9 µg of FLAG-MBP, 0.3 µg of pANAP and 0.4 µg of DN-eRF1. For experiments in which DN-eRF1 was not used, it was replaced with 0.4 µg pcDNA3. The DNA/Lipofectamine mix was prepared in 300 µL Opti-MEM (Invitrogen, Carlsbad, CA) per well. For transfection, cells were incubated in growth medium without antibiotics for 4–6 hr at 37°C with 5% $CO_2$. After incubation, the medium was replaced with one including antibiotics and supplemented with 20 µM L-Anap-ME (AssisChem, Waltham, MA). The L-Anap-ME was made as a 10 mM stock in ethanol and stored at −20°C. Trays were wrapped with aluminum foil to block any light and incubated at 37°C until use.

For fluorometry or western blot experiments, cells were harvested approximately 24 hr after transfection. Cells were washed twice with PBS, and cell pellets, collected from 9 wells each, were stored at −20°C until use. For imaging experiments, the day after transfection, medium was removed from the wells and replaced with HEPES buffered Ringers (HBR) solution (in mM: NaCl 140; KCl 4; $CaCl_2$ 1.8; glucose 5; HEPES 10; pH 7.4). Cells were then placed on the bench top at room temperature for the remainder of the same day.

## Protein purification and western blot analysis

For fluorometry experiments, between 1 and 4 frozen pellets were thawed and resuspended in 0.5–0.8 mL Stabilization Buffer Tris (SBT) (in mM: KCl 70; $MgCl_2$ 1; Trizma Base 30; pH 7.4) supplemented with cOmplete mini EDTA-free protease inhibitor cocktail (Sigma-Aldrich, St. Louis, MO). The suspension was sonicated using a Sonifier 450 with MicroTip (Branson, Danbury, CT) with settings of power = 4 and duty cycle = 50% for a total of 10 pulses. Lysed cells were then spun in a benchtop centrifuge at 13,000 rpm for 20 min at 4°C, and the cleared lysate was moved to a new tube.

Anti-FLAG M2 affinity gel (Sigma-Aldrich, St. Louis, MO) was prepared by rinsing 100 µL of slurry per sample five times with 1 mL SBT. Cleared lysate was then added to the rinsed gel and nutated at 4°C for 1 to 2 hr. Tubes were wrapped in aluminum foil to prevent photodamage. The gel was rinsed five times with 1 mL of SBT. Between 0.25 to 0.5 mL of a 200 ng/mL solution of FLAG peptide (Sigma-Aldrich, St. Louis, MO) in SBT was added to the rinsed beads, which were then nutated at 4°C for between 90 min and 12 hr to elute the protein. The purified protein was harvested by spinning the beads and collecting the supernatant. The protein was stored in tubes wrapped in aluminum foil at 4°C for up to 1 week.

For western blot analysis, 25 µL of lysis buffer per 10 mg cells were used to normalize the protein concentration for each experiment. Lysis buffer was made as follows: to 35 ml $H_2O$, 185.7 mg triethanolamine, 876.6 mg NaCl, and 10 mL glycerol were added, then the pH was adjusted to 8.0 with NaOH. After bringing the volume up to 50 mL, 500 mg digitonin, 5 mM EDTA, and Halt protease inhibitor cocktail (ThermoFisher Scientific, Waltham, MA) were added. Cleared cell lysates were then were run on NuPage 10% bis-tris gels (ThermoFisher Scientific, Waltham, MA) with MOPS SDS running buffer (50 mM MOPS, 50 mM Tris, 3.5 mM SDS and 1 mM EDTA). Proteins were transferred to PVDF membranes using a BioRad Transblot SD (Hercules, CA) transfer cell with Bjerrum/Schafer-

Nielsen transfer buffer with SDS (*Bjerrum and Schafer-Nielsen, 1986*). Membranes were blocked in 5% Milkman Instant Low Fat Dry Milk (Marron Foods, Harrison, NY) in TBS-T (20 mM Tris, 137 mM NaCl, 0.1% Tween20, pH 7.6) for either 1 hr at room temperature or overnight at 4°C. Anti-FLAG primary antibody (Sigma-Aldrich Cat F3165, St. Louis, MO) was used at a dilution of 1:20,000 and secondary antibody (Amersham Cat NA931, Pittsburgh, PA) HRP-linked anti mouse IgG was used at 1:30,000 dilution in TBS-T. Secondary antibodies were visualized with Super Signal West Femto Substrate (ThermoFisher, Waltham, MA) and imaged with a Proteinsimple gel imager (San Jose, CA). Densitometry was performed using ImageJ (*Schneider et al., 2012*). Regions of interest were drawn around each band and paired with adjacent background regions of the same size. The mean gray value within each background region was subtracted from the mean gray value within its corresponding band. We attempted to achieve similar intensities for wild-type and L-Anap incorporating MBP by loading different amounts of lysate for the different constructs. In all cases, the dilution of samples was taking into account after densitometry to allow comparisons to be made among lanes.

## Fluorometry and spectrophotometry

Starna (Atascadero, CA) sub-micro fluorometer cells (100 μL) were used for both fluorometry and spectrophotometry. Fluorometry experiments were performed using a Jobin Yvon Horiba Fluoro-Max-3 spectrofluorometer (Edison, NJ). For emission spectra of L-Anap, we used an excitation wavelength of 370 nm and 5 nm slits for excitation and emission, except for experiments to measure quantum yield in which 1 nm slits were used. For time course measurements, we excited samples at 370 nm and recorded the emission at 480 nm every 10 s using anti-photobleaching mode of the instrument. Reagents (transition metals, $Cu^{2+}$-TETAC, and DTT) were added manually as 100x stocks during the interval between measurements by pipetting up and down in the cuvette without removing it from the instrument.

Protein samples were diluted 1:10 to 1:200 in SBT to keep the fluorescence intensity within the linear range of the spectrofluorometer. We found that it was essential to use Tris buffered solutions for experiments with $Cu^{2+}$ and $Cu^{2+}$-TETAC, as $Cu^{2+}$ appeared to precipitate in HEPES-based solutions (data not shown). Transition metal ions ($Cu^{2+}$, $Co^{2+}$ and $Ni^{2+}$) were prepared from sulfate ($CuSO_4$) or chloride ($CoCl_2$ and $NiCl_2$) salts as 100 mM stocks in water or SBT, with stocks of lower concentrations diluted from this stock in SBT. We noted no difference between $CuSO_4$ and $CuCl_2$ binding to TETAC, but chose to use $CuSO_4$. TETAC (Toronto Research Chemicals, Toronto, Canada) was prepared as a 100 mM stock in DMSO and stored at $-20°C$ until the day of use. The TETAC stock appeared stable over a time scale of months and tolerated multiple freeze-thaw cycles. To prepare $Cu^{2+}$-TETAC, 1 μL each of 100 mM TETAC stock and 110 mM $CuSO_4$ stock were mixed together and allowed to incubate for one minute as the solution turned a deeper shade of blue, reflecting binding of $Cu^{2+}$ to the cyclen ring. To this mixture, 98 μL of SBT was added, giving a solution of 1.1 mM $Cu^{2+}$ and 1 mM TETAC. The high concentrations in the binding reaction and 10% over-abundance of $Cu^{2+}$ ensured that all the TETAC was bound with $Cu^{2+}$. This stock solution was then diluted 1:100 when added to the cuvette for fluorometer experiments, giving a final concentration of 10 μM $Cu^{2+}$-TETAC. $Co^{2+}$ did not appear to be compatible with TETAC, eliminating its cysteine reactivity. 1,4-Dithiothreitol (DTT) (Sigma-Aldrich, St. Louis, MO) was prepared as a 100 mM stock in water.

Absorption measurements were made using a Beckman Coulter DU 800 (Brea, CA). To measure the absorption of $Cu^{2+}$ and $Co^{2+}$ bound to HH, we used an α-helical peptide with the following sequence: Ac-ACAAKHAAKHAAAAKA-NH$_2$, which was custom synthesized by Sigma-Aldrich (St. Louis, MO) (*Taraska et al., 2009b*). The peptide was dissolved in SBT to a final concentration of 1 mM. N-ethyl maleimide was prepared as a 200 mM stock in DMSO and added to the peptide solution at a final concentration of 2 mM to prevent the peptide from precipitating upon addition of $Cu^{2+}$. To determine the extinction coefficient of $Cu^{2+}$-TETAC, we measured the absorption spectrum of 2 mM $Cu^{2+}$-cyclen.

## Calculation of quantum yield

Because the emission spectrum of L-Anap is sensitive to its environment, we measured the emission spectrum and quantum yield for L-Anap incorporated into each of our two sites in MBP. The emission spectrum of MBP-322Anap was blue shifted relative to that of MBP-295Anap by about 20 nm,

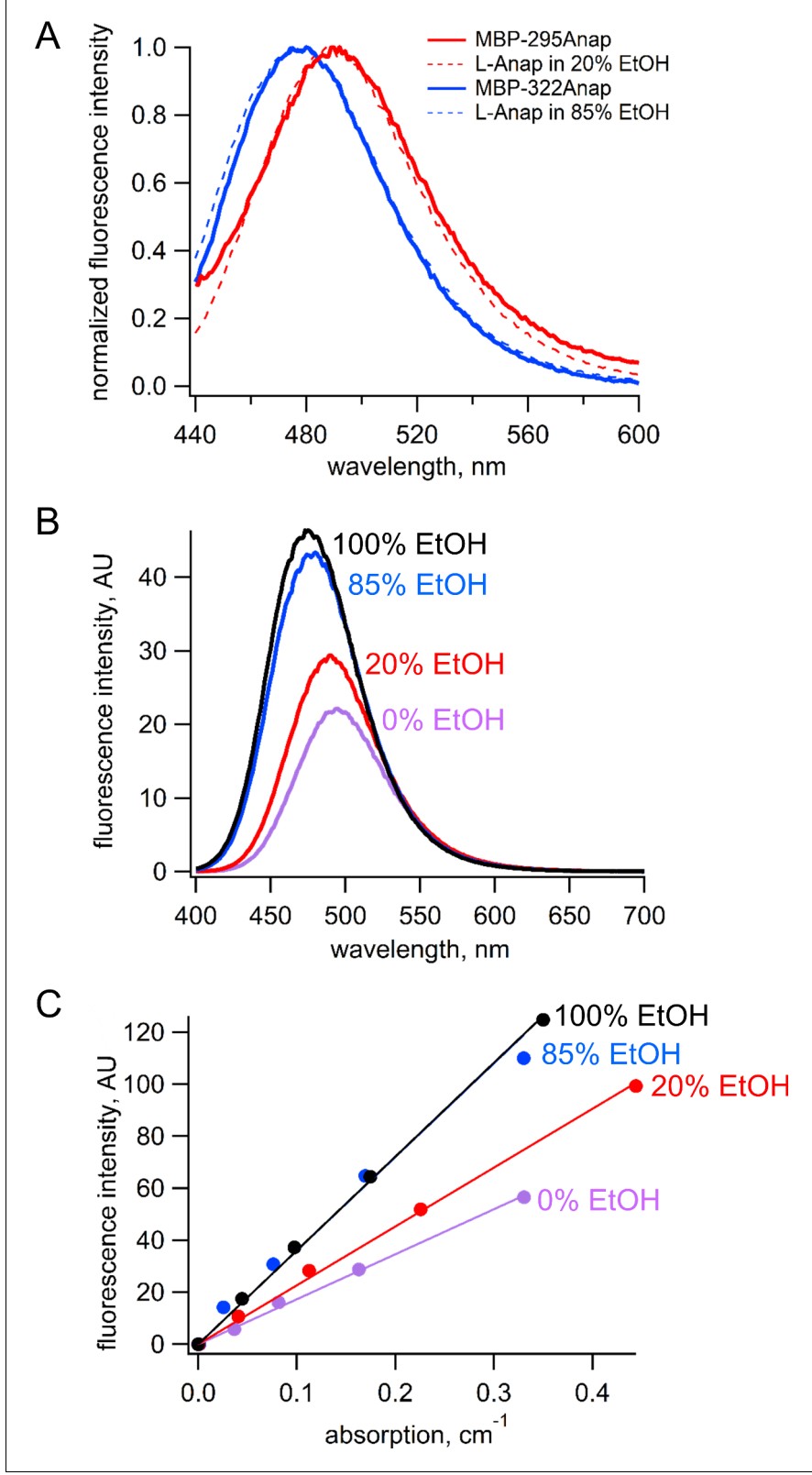

**Figure 13.** Estimation of quantum yield of Anap incorporate at the 295 and 322 sites in MBP. (**A**) Normalized emission spectra of MBP-295Anap (red line) and MBP-322Anap (blue line) showing the emission of Anap at 322 is blue shifted relative to that of Anap at 295. Also shown are the normalized emission spectra of free L-Anap in 20% ethanol (dashed red line), closely matching Anap at 295, and 85% ethanol (dashed blue line), closely matching

*Figure 13 continued on next page*

*Figure 13 continued*

Anap at 322. (**B**) Emission spectra of free L-Anap in different ethanol concentrations showing the environmental sensitivity of L-Anap. Over this range, as the environment becomes more hydrophobic, the peak fluorescence becomes larger and blue shifted. (**C**) Plot of fluorescence intensity versus absorption of free L-Anap in different ethanol concentrations. The quantum yield, calculated using the slope of these plots, increases with increasing ethanol concentration. The quantum yield of free L-Anap in 20% and 85% ethanol were used to estimate the quantum yield of Anap incorporate at the 295 and 322 sites, respectively.

DOI: https://doi.org/10.7554/eLife.37248.018

suggesting that the L-Anap in MBP-322 is in a somewhat more hydrophobic environment (*Figure 13A*). To measure the quantum yield for each L-Anap position in MBP, we identified a solvent condition for free L-Anap that would mimic the environment of the fluorophores at each of the two sites. We measured the emission spectra of free L-Anap in SBT/ethanol mixtures that ranged from 0% to 100% ethanol. As shown in *Figure 13B*, both the peak wavelength and amplitude of the emission spectra were sensitive to ethanol. We found that the emission spectrum of MBP-295TAG matched most closely to the emissions spectrum of free L-Anap in 20% ethanol (*Figure 13A*, red), whereas the emission spectrum of MBP-322Anap matched most closely to free L-Anap in 85% ethanol (*Figure 13A*, blue).

Next, we measure the quantum yield of L-Anap in the different SBT/ethanol mixtures. Samples of L-Anap were prepared in SBT, 20% ethanol in SBT, 85% ethanol in SBT, and 100% ethanol. The fluorescence intensity (excitation at 370 nm and emission at 480 nm) and absorption (at 370 nm) of each sample were measured as described above and plotted against each other in *Figure 13C*. The quantum yield of L-ANAP in each of the mixtures was calculated relative to the reference quantum yield of L-ANAP in ethanol (0.48) (*Chatterjee et al., 2013*) using the following equation (31):

$$Q_M = Q_{EtOH} \frac{slope_M}{slope_{EtOH}} \frac{\eta_M{}^2}{\eta_{EtOH}{}^2}$$

where $Q$ is the quantum yield of L-Anap, slope refers to the slope of the linear fits to the data, and $\eta$ is the refractive index. Quantum yield values were 0.23, 0.31, and 0.47 for 0%, 20%, and 85% ethanol, respectively. We therefore estimated the quantum yield for MBP-295Anap of 0.31 and for MBP-322Anap of 0.47. These estimates assume that EtOH:SBT mixtures mimic the L-Anap environment in MBP and that there are no endogenous quenchers within MBP.

## FRET efficiency analysis

For each time course experiment in the fluorometer, an averaged background trace from six to eight experiments that did not contain protein, but to which $Cu^{2+}$-TETAC and DTT, or $Cu^{2+}$ and EDTA, were added, was subtracted from the protein-containing trace. The fraction of fluorescence quenching (F) was defined as follows:

$$F = \frac{Fluoresence\ with\ metal}{Fluorescence\ without\ metal}$$

To determine the FRET efficiency, E, we corrected for other sources of energy transfer (e.g. solution quenching) using the following equations:

$$E = 1 - \frac{1}{1 + \frac{1}{F_{Cys}} - \frac{1}{F_{no\ Cys}}} \quad \text{or} \quad E = 1 - \frac{1}{1 + \frac{1}{F_{HH}} - \frac{1}{F_{no\ HH}}},$$

where $F_{Cys}$ and $F_{no\ Cys}$ are the fractional quenching by $Cu^{2+}$-TETAC in protein with and without cysteines respectively, and $F_{HH}$ and $F_{no\ HH}$ are the fraction of quenching by $Cu^{2+}$ in protein with and without HH sites, respectively. We also analyzed our data using simplified equations, which would account for nonspecific decreases in fluorescence (e.g. bleaching or loss of protein) but not energy transfer:

$$E = 1 - \frac{F_{Cys}}{F_{no\ Cys}} \quad \text{or} \quad E = 1 - \frac{F_{HH}}{F_{no\ HH}}.$$

The values of E produced by the two analysis methods were similar because of the low degree of background quenching. For clarity, only the results of analysis with the latter equations are shown. To calculate the mean and standard error of the mean for E, we used the mean and standard error of the mean for our F measurements (i.e. $F_{HH}$, $F_{No\ HH}$, $F_{Cys}$, and $F_{No\ Cys}$) in Monte Carlo resampling ($1 \times 10^6$ cycles; NIST Uncertainty Machine v1.3.4; [*Lafarge and Possolo, 2015*]).

## Imaging and unroofing

Cells were imaged and unroofed as previously described (*Gordon et al., 2016*; *Zagotta et al., 2016*). Briefly, coverslips were mounted in a homemade chamber on a microscope stage and perfused for several minutes with HBR via gravity-flow perfusion. Poly-lysine solution (0.1 mg/mL 30,000–70,000 MW in PBS) was then added to the chamber and the cells were incubated for 15 s. A 1:3 dilution of SBT was then added to the chamber, and the cells were incubated for 30 s. The solution was then replaced with SBT and the cells immediately exposed to a single pulse from a Branson Sonifier 450 with MicroTip with power = 2 and duty cycle = 50%. Unroofed cells were then perfused with SBT for at least 2 min to remove debris from the chamber before images were collected.

Experiments were performed using a Nikon Ti-E inverted microscope (Melville, NY) and a Nikon CFI Apo TIRF 60x oil immersion objective. Excitation light was provided by a Xenon arc lamp (model Lambda LS, Sutter Instrument Co., Novato, CA). For L-Anap excitation we used a 375/28 nm excitation filter and a 480/50 nm emission filter. Each experiment commenced with a pre-bleaching step, comprised of a 7-s exposure of the coverslip to L-Anap excitation light. Pre-bleaching has been previously shown to reduce small, non-specific L-Anap dependent background fluorescence (*Gordon et al., 2016*). Images for analysis were collected using exposures ranging from 100 ms to 500 ms, depending on sample intensity, using a QuantEM EMCCD camera (Photometrics, Tuscon, AZ) with readout speed of 5 MHz and the multiplier set to 20. The illumination and camera settings were the same for all fluorescence images acquired for a given field of view. Different metal concentrations and EDTA were applied via gravity perfusion. $Cu^{2+}$-TETAC and DTT were applied directly to the chamber via a transfer pipette. Five times the chamber volume was added in each case to ensure complete changeover of the solution.

## Image analysis

Images were imported into ImageJ (*Schneider et al., 2012*) for analysis. Regions of interest were drawn by hand to include most of a given cell, but excluding any regions that appeared only partially unroofed, which tended to occur at the cell edges. For each cell, a background region was selected nearby that did not contain cells. The mean gray value of the background region of interest was subtracted from the mean gray value of the region of interest of the corresponding cell. This background subtraction was repeated for each image collected in an experiment using the same regions of interest. The mean and standard error of the mean for the FRET efficiency were calculated as described above.

## Distance calculations

The $R_0$ values for each L-Anap site paired with each type of bound metal (i.e. $Cu^{2+}$ or $Cu^{2+}$-TETAC) were calculated using the measured emission spectrum and quantum yield of L-Anap at each site and the measured absorption spectra for each type of bound metal using the following equation (*Lakowicz, 2006*):

$$R_0 = C \sqrt[6]{(JQ\eta^{-4}\kappa^2)}$$

where $C$ is a scaling factor, $J$ is the normalized spectral overlap of the emission of the donor and absorption of the acceptor, $Q$ is the quantum yield of L-Anap at the given site (see above), $\eta$ is the index of refraction (1.33 in our case), and $\kappa^2$ is the orientation factor, assumed to be 2/3, a reasonable assumption for an isotropic acceptor (15). Distances were calculated from the FRET measurements using the Förster equation:

$$r = R_0 \sqrt[6]{\frac{1}{E} - 1}$$

FRET efficiencies assuming a Gaussian distribution of distances between donor and acceptor,

with FWHM = 8 Å (i.e. $\sigma$=3.4) were determined by numerically convolving the Förster equation with the Gaussian function in Microsoft (Redmond, WA) Excel 2016. The corrected distances were then determined from plots of the FRET efficiency vs. the mean distance of the Gaussian distribution.

## Acknowledgements

Research reported in this publication was supported by the National Eye Institute of the National Institutes of Health under award numbers R01EY017564 (to SEG) and R01EY010329 (to WNZ), by the National Institute of Mental Health of the National Institutes of Health under award number R01MH102378 (to WNZ), by the National Institute of General Medical Sciences of the National Institutes of Health under award numbers R01GM100718 and R01GM125351 (to SEG and WNZ), and by the following additional awards from the National Institutes of Health: S10RR025429, P30DK017047, and P30EY001730. The authors declare no competing financial interests.

## Additional information

### Funding

| Funder | Grant reference number | Author |
| --- | --- | --- |
| National Eye Institute | R01EY017564 | Sharona E Gordon |
| National Institute of General Medical Sciences | R01GM125351 | Sharona E Gordon William N Zagotta |
| National Institute of General Medical Sciences | R01GM100718 | Sharona E Gordon William N Zagotta |
| National Institutes of Health | S10RR025429 | Sharona E Gordon William N Zagotta |
| National Eye Institute | P30EY001730 | Sharona E Gordon William N Zagotta |
| National Institute of Diabetes and Digestive and Kidney Diseases | P30DK017047 | Sharona E Gordon |
| National Institute of Mental Health | R01MH102378 | William N Zagotta |
| National Eye Institute | R01EY010329 | William N Zagotta |

The funders had no role in study design, data collection and interpretation, or the decision to submit the work for publication.

### Author contributions

Sharona E Gordon, William N Zagotta, Conceptualization, Formal analysis, Funding acquisition, Investigation, Methodology, Writing—original draft, Project administration, Writing—review and editing; Mika Munari, Investigation, Writing—original draft

### Author ORCIDs

Sharona E Gordon http://orcid.org/0000-0002-0914-3361
Mika Munari http://orcid.org/0000-0003-3598-6116
William N Zagotta http://orcid.org/0000-0002-7631-8168

### Decision letter and Author response

Decision letter https://doi.org/10.7554/eLife.37248.021
Author response https://doi.org/10.7554/eLife.37248.022

## Additional files

### Supplementary files

• Transparent reporting form
DOI: https://doi.org/10.7554/eLife.37248.019

### Data availability

Data generated or analysed during this study are included in the manuscript and supporting files.

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
