## [Decision Letter]

Thank you for submitting your article "Visualizing conformational dynamics of proteins in solution and at the cell membrane" for consideration by *eLife*. Your article has been reviewed by 3 peer reviewers, including Leon D Islas as Reviewing Editor, and the evaluation has been overseen by Richard Aldrich as the Senior Editor. The following individual involved in review of your submission has also agreed to reveal their identity: Jon T Sack (Reviewer #2).

The reviewers have discussed the reviews with one another and the Reviewing Editor has drafted this decision to help you prepare a revised submission.

Summary:

This manuscript by Gordon et al. presents an improved technique to measure distances and changes in distances in soluble and membrane-bound proteins. The technique is based in the transition metal (TM) FRET technique developed by the same group. FRET-based methods are very useful, but difficult to apply to measure small distances. TM-FRET is similar to lanthanide-based FRET methods (LRET) in which the orientation dependence of the acceptor is reduced and smaller distances can be resolved. The present technique represents an advance and has the potential to be widely used. The reviewers commend the authors on a thorough approach, with many controls that support the conclusions of the manuscript.

Essential revisions:

The reviewer's main concerns are:

1) While it is mentioned in the Discussion (eighth paragraph) that the possibility of incomplete labeling with the acceptor metal ion is a limitation, it is difficult to diagnose how incomplete the labeling is in any given experiment. This is quite important, as the transformation of fluorescence to distance rests on the assumption of 100% labeling. If cysteines have been postranslationally modified (e.g. oxidation to sulfinate or sulfonate), or are sterically blocked, they could be unreactive to the copper ligand. As the FRET in unroofed cells was systematically less than with protein in solution, this seems a consideration that should be given more credence. Are there any controls that could assess the degree of incomplete labeling? Perhaps post hoc labeling with a fluorophore or some kind of superquencher could do the trick.

2) It is not clear what are the measurements used in Figure 12 to argue for the FCG. There are only two positions that were tested with two acceptors, TETAC and HH, that makes at most four points. The figure also combines measurements from soluble and membrane bound MBP. Although the idea behind the correction of the FRET curve taking into account the dynamics is probably correct, I think that to make a strong argument for it, one should make ACCuRET measurements in the same protein from pairs at many positions. The problem is actually more complicated, because one would expect different dynamics (RMSD) for each pair and even different kappa square values.

3) While the authors suggest that the use of TETAC to modify cysteines should be barely affected by unspecific modification of other (non-interest) cysteines or cysteines located farther than ~20 angstroms from the donor, this is not immediately evident and can be a big problem for membrane proteins, since TETAC will not only modify other proteins but one would expect unspecific binding to the membrane. The authors should at least acknowledge and discuss these potential problems.

4) Figure 2 describes site-specific labeling with Anap. Described data seem to be incomplete, e.g. it is not clear how the amount of full length protein was quantified; how many samples were analyzed; WB contains no loading control so it is not easy to interpret differences between the lanes (just by looking at panels B and C there is a difference in the level of WT protein), etc.

- Control for non-specific incorporation of Anap seems to be missing. Have the authors done experiments without tRNA and aa-tRNA-synthetase in the presence of Anap? In the eighth paragraph of the Discussion, they say that using amber codon suppression largely eliminates nonspecific labeling with donor, but I do not understand what are they referring to – do they see background labeling and where does it come from?

---

## [Author Response]

1) While it is mentioned in the Discussion (eighth paragraph) that the possibility of incomplete labeling with the acceptor metal ion is a limitation, it is difficult to diagnose how incomplete the labeling is in any given experiment. This is quite important, as the transformation of fluorescence to distance rests on the assumption of 100% labeling. If cysteines have been postranslationally modified (e.g. oxidation to sulfinate or sulfonate), or are sterically blocked, they could be unreactive to the copper ligand. As the FRET in unroofed cells was systematically less than with protein in solution, this seems a consideration that should be given more credence. Are there any controls that could assess the degree of incomplete labeling? Perhaps post hoc labeling with a fluorophore or some kind of superquencher could do the trick.

We agree that incomplete labeling is an important consideration for accurate measurements of distance. However we don’t believe that unreacted cysteines are a likely source of incomplete labeling for the following reasons: 1) we have measured the time course of labeling (in both the fluorometer (Figure 6) and microscope (Figure 10)) and shown that it is at steady state; 2) the labeling of HH with free Cu^2+^ gave very similar distance measurements to the labeling of cysteine with Cu^2+^-TETAC (Figure 8); and 3) the quenching in MBP-295C with maltose in the fluorometer was greater than 80%, and the 20% unquenched that remains is expected from the known distance and R_0_. Post-translational modifications of cysteine is not likely to explain the lower FRET seen in unroofed cells as the lower FRET is also observed with free Cu^2+^ in HH constructs. It seems more likely it is due to a small amount of background fluorescence in unroofed cells. We now state this more clearly in the manuscript.

2) It is not clear what are the measurements used in Figure 12 to argue for the FCG. There are only two positions that were tested with two acceptors, TETAC and HH, that makes at most four points. The figure also combines measurements from soluble and membrane bound MBP. Although the idea behind the correction of the FRET curve taking into account the dynamics is probably correct, I think that to make a strong argument for it, one should make ACCuRET measurements in the same protein from pairs at many positions. The problem is actually more complicated, because one would expect different dynamics (RMSD) for each pair and even different kappa square values.

Figure 12 shows 8 points (two donor/acceptor pairs, TETAC and HH, and with and without maltose) for the predicted distance dependence in fluorometer experiments (filled symbols). In every case, the FRET efficiency for distances longer than R_0_ is larger than the Förster prediction, and the FRET efficiency for distances less than R_0_ is smaller than the Förster prediction. This observation, seen previously in other experimental and theoretical FRET studies, is consistent with the nonlinearity of FRET and the known heterogeneity in intramolecular distances in proteins. Here we show that we can largely accounted for this decreased distance dependence use our simple FCG model (Figure 12A colored curves). Our FCG model is not intended to be the ultimate solution to this problem, and we agree that different FRET pairs would have different RMSDs and more FRET pairs would help. Nevertheless, our FCG model with an RMSD of 8 Å (typical from DEER studies) provides a better fit to our distance dependence than the Förster equation alone, which assumes and RMSD of 0 Å. Since all FRET measurements are going to involve some heterogeneity, we believe that some correction for the heterogeneity is better than no correction.

3) While the authors suggest that the use of TETAC to modify cysteines should be barely affected by unspecific modification of other (non-interest) cysteines or cysteines located farther than ~20 angstroms from the donor, this is not immediately evident and can be a big problem for membrane proteins, since TETAC will not only modify other proteins but one would expect unspecific binding to the membrane. The authors should at least acknowledge and discuss these potential problems.

We agree that endogenous cysteines can be a problem. However, this problem can be largely mitigated by: 1) selective mutation of nearby reactive endogenous cysteines and 2) correcting the FRET efficiency with the observed FRET in constructs without an introduced cysteine (no Cys). In cases where the no Cys construct reveals problematic cysteines, they can be mutated if they produce appreciable FRET or changes in FRET. Otherwise, the FRET efficiency can be corrected for these background sources using the methods described in this paper. We now elaborate more on this in the text.

4) Figure 2 describes site-specific labeling with Anap. Described data seem to be incomplete, e.g. it is not clear how the amount of full length protein was quantified; how many samples were analyzed; WB contains no loading control so it is not easy to interpret differences between the lanes (just by looking at panels B and C there is a difference in the level of WT protein), etc.

We have now expanded upon our description of the experiments in Figure 2. Densitometry analysis of the gels is now described in the Materials and methods, and the number of samples is in the figure legend. Although in retrospect loading controls would have been useful, the samples were instead normalized to the mass of the cell pellet, with 25 µL of lysis buffer used per 10 mg of cells. We have improved our description of this process in the Materials and methods section. The darker band for wt MBP observed by the reviewer in Figure 2C compared to Figure 2B is due to a different amount of lysate loaded for the two gels (8 x more in Figure 2B than 2C).

- Control for non-specific incorporation of Anap seems to be missing. Have the authors done experiments without tRNA and aa-tRNA-synthetase in the presence of Anap? In the eighth paragraph of the Discussion, they say that using amber codon suppression largely eliminates nonspecific labeling with donor, but I do not understand what are they referring to – do they see background labeling and where does it come from?

Figure 2—figure supplement 1 shows a control for the nonspecific incorporation of Anap. Here we show that MBP is the only detectable band with Anap fluorescence, even in the crude cell lysates. We have expanded our discussion of these data in the Results and Discussion. We also explain that the use of amber codon suppression substantially reduces nonspecific labeling relative to using cysteine-reactive donors for labeling.